Journal of Data-centric Machine Learning Research (2024)   Submitted 11/23; Revised 06/24; Published 07/24

# Datasets and Benchmarks for Offline Safe Reinforcement Learning

**Zuxin Liu**[1*]                                                    ZUXINL@ANDREW.CMU.EDU
**Zijian Guo**[1*]                                                    ZIJIANG@ANDREW.CMU.EDU
**Haohong Lin**[1]                                                 HAOHONGL@ANDREW.CMU.EDU
**Yihang Yao**[1]                                                   YIHANGYA@ANDREW.CMU.EDU
**Jiacheng Zhu**[1]                                                      JZHU4@ANDREW.CMU.EDU
**Zhepeng Cen**[1]                                                      ZCEN@ANDREW.CMU.EDU
**Hanjiang Hu**[1]                                               HANJIANGH@ANDREW.CMU.EDU
**Wenhao Yu**[2]                                                  MAGICMELON@GOOGLE.COM
**Tingnan Zhang**[2]                                                   TINGNAN@GOOGLE.COM
**Jie Tan**[2]                                                          JIETAN@DEEPMIND.COM
**Ding Zhao**[1]                                                         DINGZHAO@CMU.EDU
[1]*Carnegie Mellon University*, [2]*Google Deepmind*

Reviewed on OpenReview: HTTPS://OPENREVIEW.NET/FORUM?ID=AJF5PE3Z7W

**Editor:** Jakob Nicolaus Foerster

## Abstract

This paper presents a comprehensive benchmarking suite tailored to offline safe reinforcement learning (RL) challenges, aiming to foster progress in the development and evaluation of safe learning algorithms in both the training and deployment phases. Our benchmark suite contains three packages: 1) expertly crafted safe policies, 2) D4RL-styled datasets along with environment wrappers, and 3) high-quality offline safe RL baseline implementations. We feature a methodical data collection pipeline powered by advanced safe RL algorithms, which facilitates the generation of diverse datasets across 38 popular safe RL tasks, from robot control to autonomous driving. We further introduce an array of data post-processing filters, capable of modifying each dataset's diversity, thereby simulating various data collection conditions. Additionally, we provide elegant and extensible implementations of prevalent offline safe RL algorithms to accelerate research in this area. Through extensive experiments with over 50000 CPU and 800 GPU hours of computations, we evaluate and compare the performance of these baseline algorithms on the collected datasets, offering insights into their strengths, limitations, and potential areas of improvement. Our benchmarking framework serves as a valuable resource for researchers and practitioners, facilitating the development of more robust and reliable offline safe RL solutions in safety-critical applications. The benchmark website is available at `www.offline-saferl.org`.

**Keywords:**  benchmark, offline RL, safe RL

---

∗. equal contribution, corresponding to `zuxinl@andrew.cmu.edu`

# 1 Introduction

Reinforcement learning (RL) has shown remarkable success in a broad array of domains, from game playing to robot control (Mnih et al., 2015; Ibarz et al., 2021). Nevertheless, a paramount challenge persists: guaranteeing safety throughout both the training and deployment phases (Gu et al., 2022). This is of particular concern in applications where unsafe behaviors could lead to catastrophic outcomes (Xu et al., 2022b). Traditional online RL often struggles to optimally balance safety performance and task performance, frequently leading to policies that are either excessively risky or unduly conservative. This is because of the difficulty in integrating cost and reward signals into a singular objective function without sidelining either aspect. Unlike traditional methods that rely on fixed coefficients for cost terms, safe RL dynamically adjusts these coefficients in response to the policy's current risk profile (Chen et al., 2021a; Yao et al., 2023; Liu et al., 2023a), offering a more adaptable solution to enforce safety constraints. However, the requisite online interactions for this dynamic adjustment pose significant risks during the training phase, particularly in safety-critical applications like autonomous driving, where the exploration of unknown actions on real roads is unfeasible (Lin et al., 2023).

Given these considerations, offline safe learning emerges as a pivotal research area, focusing on the development of constrained policies from pre-collected datasets that ensure safety throughout the learning process (Levine et al., 2020). These approaches sidestep the dangers associated with online exploration, thereby enhancing the practicality of RL applications in safety-sensitive environments.

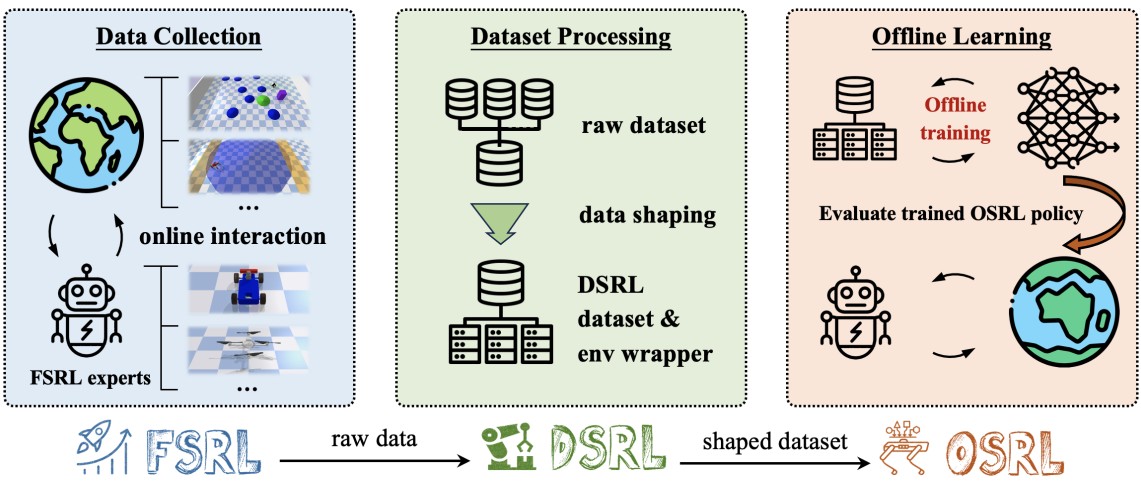

Figure 1: Overview of the benchmark with three packages: `FSRL`, `DSRL`, and `OSRL`.

Despite the rising significance of offline learning, public benchmarks or datasets specifically designed to address the safety aspect are notably scarce. Conventional datasets, like D4RL (Fu et al., 2020), are excellent proving grounds for traditional offline learning algorithms, but their primary objective is reward maximization without any explicit safety constraints (Seno and Imai, 2022). This notable gap impedes the progress of deploying RL safely to real-world applications. There is a clear need for a specialized benchmark and dataset to train, evaluate, and compare safe learning algorithms under constraints.

To address this gap, we introduce a comprehensive benchmarking platform and datasets for offline safe learning, comprised of three packages: FSRL (Fast Safe RL), DSRL (Datasets for Safe RL), and OSRL (Offline Safe RL) as shown in Figure 1. FSRL incorporates an efficient data collection pipeline with parallel workers from Tianshou (Weng et al., 2021) to generate datasets suitable for safety-embedded tasks. We provide 38 high-quality datasets across different difficulty levels in three widely-used safe RL environments: Mujoco-based Safety-Gymnasium (Ray et al., 2019; Ji et al., 2023b), PyBullet-based BulletSafetyGym (Gronauer, 2022), and Panda3D-based self-driving simulator MetaDrive (Li et al., 2022).

DSRL hosts these offline datasets, offering a consistent API with D4RL, which is commonly used in offline RL, for easy usage and online evaluation of offline learning methods (Fu et al., 2020). We also furnish an array of deterministic data post-processing filters that can alter data density, noise level, as well as the distributions of rewards and costs, simulating diverse data collection conditions. This produces hundreds of distinct datasets of varying difficulty levels. Importantly, our framework doesn't only supply pre-collected datasets but also establishes a systematic approach to data collection and processing, enabling easy extension to other domains for future datasets and fostering a continually evolving benchmarking ecosystem.

Furthermore, we offer the OSRL codebase, implementing a broad spectrum of existing offline safe RL algorithms (Xu et al., 2022a; Lee et al., 2022; Liu et al., 2023b) for offline learning and their corresponding online evaluation scripts. This serves as a solid foundation for the safe RL community to build upon and benchmark against. To provide insights into their strengths and limitations, we conduct an extensive empirical analysis of these baseline algorithms using our benchmark datasets.

In summary, our contributions are as follows:

- **We introduce a comprehensive benchmarking platform tailored for offline safe learning**, providing a standard testing ground for the evaluation and comparison of safe learning algorithms.

- **We offer a collection of post-processing filters to simulate diverse data collection conditions**, yielding distinct datasets with varying difficulty levels.

- **We implement the D4RL-style data wrapper and state-of-the-art offline safe learning algorithms**, serving as a good starting point for researchers and practitioners in this area.

- **We conduct a thorough empirical analysis**, utilizing over 50,000 CPU hours and 800 GPU hours of computation, providing us insights into the strengths and limitations of offline safe RL algorithms.

By making our datasets and codebase publicly available, we aim to foster collaboration, accelerate innovation, and contribute to the broader adoption of safe RL solutions in safety-critical applications.

## 2 Related Work

**Safe RL and Benchmarks.** Ensuring safety during RL training and deployment is a challenging problem (Gu et al., 2022; Xu et al., 2022b). Numerous techniques have been

explored to incorporate safety constraints into RL, such as constrained optimization (Sootla et al., 2022; Yang et al., 2021; Liu et al., 2022a), Lagrangian-based methods (Chow et al., 2017; Chen et al., 2021c), and correction-based approaches (Zhao et al., 2021; Luo and Ma, 2021). Despite these efforts, guaranteeing zero constraint violations during training is a formidable task (Dalal et al., 2018; Brunke et al., 2021). While there are benchmarks available for safe RL algorithms (Ray et al., 2019; Ji et al., 2023b) and environments (Gronauer, 2022; Ji et al., 2023a), the lack of a comprehensive suite that targets offline training remains a gap in the field.

**Offline RL and Benchmarks.** Offline RL techniques aim to learn effective policies from pre-collected data without further environment interactions (Ernst et al., 2005; Levine et al., 2020). It promises to enhance the scalability and efficiency of RL, particularly in applications where real-time interaction is expensive, risky, or impractical (Fu et al., 2019; Brandfonbrener et al., 2021). Offline RL is characterized by unique challenges, primarily arising from distributional shift, which can lead to extrapolation errors when learning policies beyond the support of the data distribution (Fujimoto et al., 2019; Kumar et al., 2020). To combat these challenges, several strategies have been proposed, such as incorporating regularization or constraints (Wu et al., 2019; Peng et al., 2019; Kostrikov et al., 2021), or leveraging techniques like importance sampling to reduce estimation variance (Nachum et al., 2019). While there are prevalent testing grounds for offline RL algorithms (Fu et al., 2020; Gulcehre et al., 2020; Tarasov et al., 2022); however, they lack explicit safety constraints in their datasets.

**Offline Safe RL.** The intersection of offline RL and safe RL has recently been a focus of attention, where techniques from both fields are leveraged (Le et al., 2019). For instance, stationary distribution correction-based methods have been used to formulate the constrained optimization problem (Lee et al., 2022; Polosky et al., 2022), while Lagrangian-based approaches have been integrated with offline RL methods to provide safe learning (Xu et al., 2022a). Sequential decision-making algorithms such as Decision Transformers have also been explored in this area (Liu et al., 2023b; Zhang et al., 2023). Despite these developments, there are no publicly safe RL datasets and algorithm libraries to compare these methods, revealing a clear need for a benchmarking framework in this vital area.

## 3 Preliminaries

### 3.1 Constrained Markov Decision Process and Safe RL

Safe RL is usually formulated under the Constrained Markov Decision Process (CMDP) framework (Altman, 1998). A finite horizon CMDP, denoted as $\mathcal{M}$, consists of a tuple $(\mathcal{S}, \mathcal{A}, \mathcal{P}, r, c, \mu_0)$, where $\mathcal{S}$ represents the state space, $\mathcal{A}$ the action space, $\mathcal{P} : \mathcal{S} \times \mathcal{A} \times \mathcal{S} \rightarrow [0,1]$ the transition function, $r : \mathcal{S} \times \mathcal{A} \times \mathcal{S} \rightarrow \mathbb{R}$ the reward function, and $\mu_0 : \mathcal{S} \rightarrow [0,1]$ the initial state distribution. In addition to these elements in a typical MDP, CMDP incorporates an extra cost function $c : \mathcal{S} \times \mathcal{A} \times \mathcal{S} \rightarrow [0, C_{max}]$ to account for constraint violations, with $C_{max}$ being the maximum cost.

A safe RL problem is specified by a CMDP and a constraint threshold $\kappa$. A policy $\pi : \mathcal{S} \times \mathcal{A} \rightarrow [0,1]$ maps the state-action space to probabilities, and a trajectory $\tau = \{s_1, a_1, r_1, c_1..., s_T, a_T, r_T, c_T\}$ contains state and action, reward, and cost information throughout the maximum episode length $T$. The cumulative reward and cost for a

trajectory $\tau$ are represented as $R(\tau) = \sum_{t=1}^{T} r_t$ and $C(\tau) = \sum_{t=1}^{T} c_t$, respectively. Safe RL aims to find a policy that maximizes the reward while keeping the constraint violation cost below the threshold $\kappa$:

$$\max_{\pi} \mathbb{E}_{\tau \sim \pi}\big[R(\tau)\big], \quad s.t. \quad \mathbb{E}_{\tau \sim \pi}\big[C(\tau)\big] \leq \kappa. \tag{1}$$

The majority of literature focuses on the online setting (Achiam et al., 2017; Zhang et al., 2020; Liu et al., 2022b), where the agent is allowed to interact with the environment to gather fresh trajectory data. Conversely, in the offline setting, the agent must rely on pre-collected trajectories from unknown policies, which poses challenges for solving this constrained optimization problem.

### 3.2 Characterizing Dataset with Constraints

The cost-reward plot is a commonly used tool in safe reinforcement learning (RL), used to visualize the diversity of offline datasets by plotting the total reward and cost of each trajectory (Liu et al., 2023b). This scatter plot reveals the dataset's diversity and the trade-offs between maximizing rewards and satisfying constraints. The spread and shape of points on the plot indicate dataset complexity and the inherent challenge of balancing high-reward opportunities against their risks (Liu et al., 2022b). We aim to enhance trajectory diversity by covering the cost-reward metric space extensively. While trajectories may share similar cost and reward outcomes, differing cost-reward returns ensure each trajectory's uniqueness. Our goal is to populate the metric space with diverse trajectories to enable the resolution of target problems within the dataset's scope. Although not all discarded trajectories might be redundant, our selection process ensures the inclusion of diverse trajectories that are sufficient for solving target problems. Therefore, the cost-reward plot serves as an intuitive tool for understanding the dataset's property, complexity, and diversity. This, in turn, aids in selecting appropriate datasets for benchmarking offline safe RL algorithms.

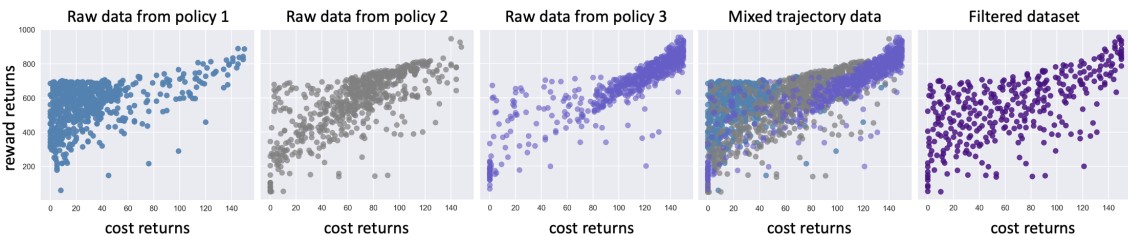

Figure 2: Illustration of the cost-reward plot and data collection from a mixture of experts. Multiple expert policies are trained, the raw data is collected, and a density filter is applied to remove redundant trajectories concentrated within the same region.

## 4 Datasets and Benchmarks

### 4.1 Dataset Collection

Our objective is to collect an array of high-quality datasets that span a spectrum of difficulty levels, thus enabling an unbiased evaluation of various algorithms' capabilities. With this

goal, we implement the `FSRL` library, containing advanced safe RL algorithms to generate a broad spectrum of datasets. The supported algorithms include the PID Lagrangian-based methods (Ray et al., 2019; Stooke et al., 2020), first-order method (Zhang et al., 2020), second-order method (Achiam et al., 2017), and variational-inference-based method (Liu et al., 2022a). Each task is trained by a suite of expert policies subjected to varying cost thresholds, thereby obtaining a pool of raw data that captures the intricacies of different task scenarios. Subsequently, we apply a density filter across the cost-reward return space. This filter removes redundant trajectories that are highly concentrated within the same region, thereby maintaining greater diversity within the dataset. The full procedure is visualized in Figure 2. More details regarding the implemented expert algorithms, training techniques, and hyperparameters are available in the supplementary material. Through this methodical approach to data collection, we strive to provide a rich and varied foundation for assessing the strengths and limitations of offline safe RL algorithms under a broad range of conditions.

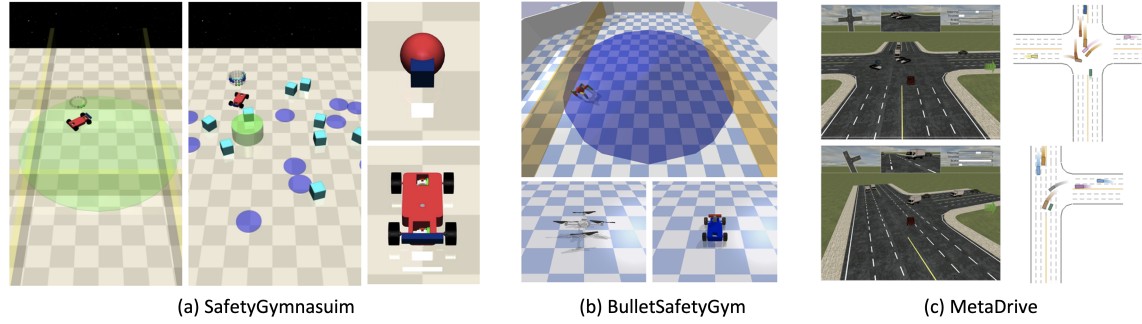

(a) SafetyGymnasuim  (b) BulletSafetyGym  (c) MetaDrive

Figure 3: Visualization of the simulation environments and representative tasks.

**Simulation Environments and Tasks.** We gather datasets from three widely recognized safe RL environments: **1) SafetyGymnasium** (Ray et al., 2019; Ji et al., 2023b), a collection of environments based on the Mujoco physics simulator, which offers a diverse range of tasks, with various safety constraints and challenges that can be adjusted to create different difficulty levels. **2) BulletSafetyGym** (Gronauer, 2022), a suite of environments built on top of the PyBullet physics simulator, which is similar to SafetyGymnasium but with shorter horizons and more agents. **3) MetaDrive** (Li et al., 2022), a self-driving simulator based on the Panda3D game engine (Goslin and Mine, 2004), which provides intricate road conditions and dynamic scenarios that closely emulate real-world driving situations, enabling the evaluation of safe RL algorithms in high-stakes, realistic applications. Figure 3 visualize some representative tasks of these environments. For example, the Safety-Gymnasium constrains navigation tasks, such as reaching goals while avoiding obstacles, and locomotion tasks, such as controlling an agent to move forward within a velocity limit. More details of the environments can be found in Appendix A.

An overview of these environments and tasks is presented in Table 1. We totally collect over 75000 diverse trajectories from 38 tasks. A detailed breakdown of these datasets, including task names, trajectory size, observation space, and action space, can be found in the supplementary material. By gathering datasets from these distinct environments,

| Benchmarks | Backends | Environments | Agents | Difficulty Levels | Total Tasks | Dataset Trajectories |
|---|---|---|---|---|---|---|
| SafetyGymnasium | Mujoco | Goal, Button, Push, Circle | Point, Car | 2 | 16 | 40310 |
| | | Velocity | Ant, HalfCheetah, Hopper, Swimmer, Walker2d | 1 | 5 | 11399 |
| BulletSafetyGym | PyBullet | Run, Circle | Ball, Car, Drone, Ant | 1 | 8 | 14498 |
| MetaDrive | Panda3D | Driving | Vehicle | 3 | 9 | 9000 |

Table 1: Overview of the safe RL benchmarks and tasks for dataset collection

we ensure a well-rounded evaluation and benchmarking process that accurately reflects the capabilities of offline safe RL algorithms across a wide spectrum of tasks and complexities.

## 4.2 Dataset Wrapper and Post-process Filters

We provide and maintain all the collected datasets via the `DSRL` package, which follows the same user-friendly API structure as D4RL (Fu et al., 2020), facilitating the usage for researchers. The key distinction lies in the inclusion of a specialized `costs` entry in the datasets for indicating constraint violations.

Apart from the access to the full datasets that are diverse over the cost-reward return space, we also provide a set of post-process filters to adjust the complexity and difficulty level of each dataset, aiming to achieve a comprehensive evaluation for different perspectives, as we will introduce in section 4.3. This idea is similar to the data augmentation technique in improving the performance and robustness of models (Mikołajczyk and Grochowski, 2018; Dao et al., 2019; Maharana et al., 2022) This involves changing data density, discarding data within specific cost-reward ranges, and increasing outlier trajectories. Here is a detailed description of these filters:

**Filter for Data Density Variation**: This filter aims to create datasets with varying data densities. These variations will help evaluate the algorithms' ability to perform under different levels of data availability and assess their generalization capabilities.

**Filter for Partial Data Discarding**: This filter operates by selectively removing trajectories within defined return ranges, represented as $[(r_{\min}, c_{\min}), (r_{\max}, c_{\max})]$. Typically, by discarding trajectories with low costs and high rewards, we alter the reward distribution, mirroring scenarios where data is collected by either overly conservative or excessively risky policies. Another us-

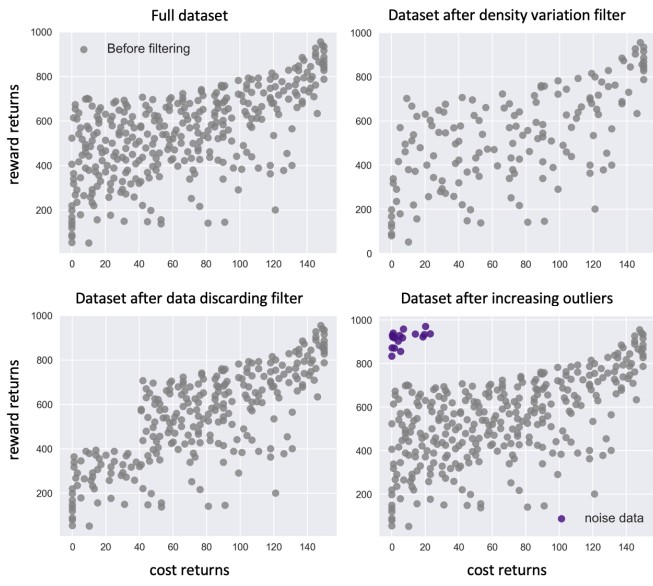

Figure 4: Illustration of the post-process filters.

age scenario is removing data within specific cost ranges, which can assess an algorithm's ability to manage unseen safety thresholds and learn from sub-optimal data.

**Filter for Noise Level Manipulation**: The real-world dataset could be noisy and contains trajectories that accidentally record high reward returns and small cost returns despite following a poor behavioral policy. We consider the task with stochastic reward and cost function, i.e., high-cost trajectories have the probability of $\alpha\%$ to be labeled as a "lucky" trajectory with high reward and low cost. The noise level manipulation filter introduces different degrees of outliers to the dataset trajectories Specifically, we select $\alpha\%$ high-cost trajectories and modify their cost return to be less than the cost threshold and their reward returns to be high. This process simulates outlier trajectories that could potentially mislead learning algorithms by promoting hazardous behaviors, mimicking scenarios where abnormality arises during data collection. With this filter in place, we assess the algorithm's capacity to manage noisy data and its tolerance towards outliers.

These filters function deterministically, ensuring the resulting datasets remain consistent given a fixed parameter, thereby maintaining fair comparison between algorithms. In addition, they offer compositional flexibility when applied to the entire dataset, permitting customization of learning problem difficulty levels. As a result, our benchmark offers hundreds of unique datasets, each embodying different challenge levels, allowing us to efficiently evaluate various aspects of safe learning algorithms across a diverse range of complexity, as we will introduce in the next section. We also present examples of using these filters in the experiments and supplementary material.

## 4.3 Evaluating Offline Safe RL Algorithms

Given the distinctive problem setting, objectives, and applications of offline safe RL compared to standard RL, we reevaluate the methodology for comparing offline safe RL approaches. Accordingly, we propose a tiered, four-level evaluation perspective, organized in order of importance, to assess safe offline reinforcement learning algorithms more accurately.

**Safety Compliance**: This primary requirement evaluates an agent's adherence to specific safety rules. The learning agent must not violate any safety constraints while striving to maximize rewards.

**Reward-seeking**: Alongside safety, we assess the agent's ability to maximize rewards within the safety set. We want to prevent the agent from being overly conservative and achieve a balance between safety and reward-seeking behavior.

While these initial levels focus on fundamental requirements, the remaining levels target more advanced attributes that add value to offline safe RL algorithms.

**Generalization**: A generalizable safe learning algorithm should be capable of learning from sub-optimal data and be robust to unseen samples upon deployment, as obtaining datasets that comprehensively cover every possible scenario is impractical. This perspective measures the agent's adaptability to previously unseen training conditions and requirements.

**Outlier Sensitivity**: This attribute tests the algorithm's resilience against outliers in the datasets. The algorithm's performance against outlier trajectories with abnormally high rewards and low costs is assessed, ensuring it can effectively handle noisy or imperfect data without compromising safety.

We present these evaluation perspectives as a guiding resource for future researchers and practitioners. While our benchmark primarily centers on the first two levels of evaluation, we supply an array of filters to post-process the datasets. These filters aid in assessing the generalization capabilities and outlier sensitivity of learning algorithms, as we presented in the previous section.

## 4.4 Offline Safe RL Benchmarks

We implemented a broad range of existing offline safe RL methods in our benchmark, all organized under the `OSRL` package, whose framework design is mainly inspired by the user-friendly CORL (Tarasov et al., 2022) library. A detailed summary of these methods is provided in Table 2. The *Type* row categorizes the learning algorithm (such as Q-learning or imitation learning), while the *Base Method* row indicates the corresponding offline learning method, excluding safety constraints considerations. As far as we know, these implemented algorithms represent the majority of the offline safe learning categories currently available in the literature.

It's worth highlighting that the Lagrangian-based methods align with the expert safe RL policy implementation in our `FSRL` package, which utilizes adaptive PID-based Lagrangian multipliers to penalize constraint violations (Stooke et al., 2020). This approach can be effortlessly extended to other existing Q-learning-based offline RL methods. In addition, we've prioritized coherence and user-friendliness in the API structure of these implementations, aiming to provide a valuable resource that can contribute to and further the development of the offline safe RL community.

| Type | Sequential Modeling | Imitation Learning | Distribution Correction Estimation | Q-learning | | |
|---|---|---|---|---|---|---|
| Algorithm | CDT (Liu et al., 2023b) | BC-{Safe, All} (Liu et al., 2023b; Xu et al., 2022a) | COptiDICE (Lee et al., 2022) | CPQ (Xu et al., 2022a) | BCQ-Lag (Xu et al., 2022a) | BEAR-Lag (Xu et al., 2022a) |
| Base Method | Decision Transformer (Chen et al., 2021b) | Behavior Cloning | OptiDICE (Lee et al., 2021) | BCQ (Fujimoto et al., 2019) | BCQ (Fujimoto et al., 2019) Lagrangian (Stooke et al., 2020) | BEAR (Kumar et al., 2019) Lagrangian (Stooke et al., 2020) |

Table 2: Implemented offline safe learning algorithms and their base methods.

## 4.5 Evaluation Metrics

We adopt the normalized reward return and the normalized cost return as the comparison metrics (Fu et al., 2020; Liu et al., 2023b). Denote $r_{\max}(\mathcal{M})$ and $r_{\min}(\mathcal{M})$ as the maximum empirical reward return and the minimum empirical reward return for task $\mathcal{M}$. The normalized reward is computed by:

$$R_{\text{normalized}} = \frac{R_\pi - r_{\min}(\mathcal{M})}{r_{\max}(\mathcal{M}) - r_{\min}(\mathcal{M})},$$

where $R_\pi$ denotes the evaluated reward return of policy $\pi$. Note that we use a constant maximum and minimum values for a safe RL task rather than a dataset. This is because the post-process filters may modify the dataset to create different difficulty levels.

The normalized cost is defined differently from the reward to better distinguish the results. It is computed by the ratio between the evaluated cost return $C_\pi$ and the target

threshold $\kappa$:

$$C_{\text{normalized}} = \frac{C_\pi + \epsilon}{\kappa + \epsilon},$$

where $\epsilon$ is a positive number to ensure numerical stability if the threshold $\kappa = 0$ and $\epsilon = 0$ if $\kappa \neq 0$. Note that the cost return and the threshold are always non-negative. Without otherwise statements, we will abbreviate "normalized cost return" as "cost" and "normalized reward return" as "reward" for simplicity.

## 5 Experiments and Analysis

### 5.1 Experiment Settings

Contrary to earlier safe RL studies, which tested agents under a single threshold, we adopt a **Constraint Variation Evaluation** to assess algorithm versatility. By training agents with varying safety constraint requirements, we can evaluate an algorithm's adaptability to a diverse range of safety conditions. This is similar to the Average Precision or the Area Under the Curve metrics in the literature (Davis and Goadrich, 2006). Note that the algorithms implemented (except CDT) need re-training from scratch for different cost thresholds which is impractical since it requires training thousands of policies. Empirically, each algorithm is trained and evaluated on each dataset using three distinct target cost thresholds and with three random seeds. These target cost thresholds range from low to high, with the lower thresholds posing a challenge for training due to limited safe trajectories and an abundance of tempting but risky ones, and the higher thresholds presenting the opposite scenario. We then compute the average of the normalized reward and cost to characterize the performance on varying safety conditions better.

**Hyperparameter Tuning Procedure:** We employ a three-step process to tune the hyperparameters. Note that we perform online evaluations to determine the performance metrics, aiming to test the maximum capabilities of these algorithms without bias. It is important to acknowledge that the best practice for hyperparameter tuning in offline RL without environment interaction remains an active area of research and falls outside the scope of this paper.

1. **Baseline Optimization:** We begin by initializing the hyperparameters with those used in Tianshou, setting the cost limit to infinity. We then fine-tune common hyperparameters such as learning rate, batch size, and hidden layer sizes to ensure the agent can complete the task without safety constraints. With minimal adjustments to Tianshou's default values, most base algorithms successfully solve the task after a few trials.

2. **Safety Constraint Integration:** Next, we introduce a cost threshold of 40 and focus on tuning safety-related parameters, such as the safety penalty coefficient for Lagrangian-based approaches. We conduct a grid search for these key hyperparameters, selecting the combination that yields the best overall performance.

3. **Algorithm-Specific Optimization:** Finally, we perform a grid search for algorithm-dependent hyperparameters. For instance, in the case of CDT (Constrained Decision Transformer), we optimize the number of attention heads.

Detailed hyperparameter configurations are provided in Appendix A.4.

Except for the experiments for CDT, which are conducted with NVIDIA A100 GPUs, all other experiments are conducted with AMD EPYC 7542 32-Core CPUs or Intel Xeon CPUs with 4 threads. The longest experiment takes approximately one day.

| Task | BC-All | | BC-Safe | | CDT | | BCQ-Lag | | BEAR-Lag | | CPQ | | COptiDICE | |
|---|---|---|---|---|---|---|---|---|---|---|---|---|---|---|
| | reward ↑ | cost ↓ | reward ↑ | cost ↓ | reward ↑ | cost ↓ | reward ↑ | cost ↓ | reward ↑ | cost ↓ | reward ↑ | cost ↓ | reward ↑ | cost ↓ |
| PointButton1 | $0.1_{\pm0.06}$ | $1.05_{\pm0.39}$ | $\mathbf{0.06_{\pm0.04}}$ | $\mathbf{0.52_{\pm0.21}}$ | $0.53_{\pm0.01}$ | $1.68_{\pm0.13}$ | $0.24_{\pm0.04}$ | $1.73_{\pm1.11}$ | $0.2_{\pm0.04}$ | $1.6_{\pm0.99}$ | $0.69_{\pm0.05}$ | $3.2_{\pm1.57}$ | $0.13_{\pm0.02}$ | $1.35_{\pm0.91}$ |
| PointButton2 | $0.27_{\pm0.08}$ | $2.02_{\pm0.38}$ | $0.16_{\pm0.04}$ | $1.1_{\pm0.84}$ | $0.46_{\pm0.01}$ | $1.57_{\pm0.1}$ | $0.4_{\pm0.03}$ | $2.66_{\pm1.47}$ | $0.43_{\pm0.05}$ | $2.47_{\pm1.17}$ | $0.58_{\pm0.07}$ | $4.3_{\pm2.35}$ | $0.15_{\pm0.03}$ | $1.51_{\pm0.96}$ |
| PointCircle1 | $0.79_{\pm0.05}$ | $3.98_{\pm0.55}$ | $\mathbf{0.41_{\pm0.08}}$ | $\mathbf{0.16_{\pm0.11}}$ | $\mathbf{0.59_{\pm0.0}}$ | $\mathbf{0.69_{\pm0.04}}$ | $0.54_{\pm0.17}$ | $2.38_{\pm1.3}$ | $0.73_{\pm0.11}$ | $3.28_{\pm2.07}$ | $\mathbf{0.43_{\pm0.07}}$ | $\mathbf{0.75_{\pm1.86}}$ | $0.86_{\pm0.01}$ | $5.51_{\pm2.93}$ |
| PointCircle2 | $0.66_{\pm0.09}$ | $4.17_{\pm0.72}$ | $\mathbf{0.48_{\pm0.08}}$ | $\mathbf{0.99_{\pm0.35}}$ | $0.64_{\pm0.01}$ | $1.05_{\pm0.08}$ | $0.66_{\pm0.13}$ | $2.6_{\pm0.71}$ | $0.63_{\pm0.27}$ | $4.27_{\pm1.48}$ | $0.24_{\pm0.4}$ | $3.58_{\pm3.09}$ | $0.85_{\pm0.01}$ | $8.61_{\pm4.62}$ |
| PointGoal1 | $0.65_{\pm0.03}$ | $0.95_{\pm0.07}$ | $0.43_{\pm0.12}$ | $0.54_{\pm0.24}$ | $0.69_{\pm0.02}$ | $1.12_{\pm0.07}$ | $\mathbf{0.71_{\pm0.02}}$ | $\mathbf{0.98_{\pm0.46}}$ | $0.74_{\pm0.02}$ | $1.18_{\pm0.64}$ | $\mathbf{0.57_{\pm0.08}}$ | $\mathbf{0.35_{\pm0.37}}$ | $0.49_{\pm0.05}$ | $1.66_{\pm1.05}$ |
| PointGoal2 | $0.54_{\pm0.03}$ | $1.97_{\pm0.24}$ | $\mathbf{0.29_{\pm0.09}}$ | $\mathbf{0.78_{\pm0.27}}$ | $0.59_{\pm0.03}$ | $1.34_{\pm0.05}$ | $0.67_{\pm0.06}$ | $3.18_{\pm1.79}$ | $0.67_{\pm0.03}$ | $3.11_{\pm1.76}$ | $0.4_{\pm0.15}$ | $1.31_{\pm0.71}$ | $0.38_{\pm0.03}$ | $1.92_{\pm1.15}$ |
| PointPush1 | $0.19_{\pm0.07}$ | $0.61_{\pm0.05}$ | $0.13_{\pm0.05}$ | $0.43_{\pm0.29}$ | $0.24_{\pm0.02}$ | $0.48_{\pm0.05}$ | $\mathbf{0.33_{\pm0.04}}$ | $\mathbf{0.86_{\pm0.45}}$ | $0.22_{\pm0.04}$ | $0.79_{\pm0.39}$ | $0.2_{\pm0.08}$ | $\mathbf{0.83_{\pm0.44}}$ | $\mathbf{0.13_{\pm0.02}}$ | $\mathbf{0.83_{\pm0.52}}$ |
| PointPush2 | $0.18_{\pm0.02}$ | $0.91_{\pm0.1}$ | $0.11_{\pm0.04}$ | $0.8_{\pm0.59}$ | $0.21_{\pm0.04}$ | $0.65_{\pm0.03}$ | $\mathbf{0.23_{\pm0.03}}$ | $\mathbf{0.99_{\pm0.57}}$ | $0.16_{\pm0.05}$ | $0.89_{\pm0.59}$ | $0.11_{\pm0.14}$ | $1.04_{\pm0.61}$ | $0.02_{\pm0.07}$ | $1.18_{\pm0.74}$ |
| CarButton1 | $0.03_{\pm0.1}$ | $1.38_{\pm0.41}$ | $\mathbf{0.07_{\pm0.03}}$ | $\mathbf{0.85_{\pm0.39}}$ | $0.21_{\pm0.02}$ | $1.6_{\pm0.12}$ | $0.04_{\pm0.05}$ | $1.63_{\pm0.59}$ | $0.18_{\pm0.05}$ | $2.72_{\pm2.23}$ | $0.42_{\pm0.05}$ | $9.66_{\pm5.71}$ | $-0.08_{\pm0.09}$ | $1.68_{\pm1.29}$ |
| CarButton2 | $-0.13_{\pm0.01}$ | $1.24_{\pm0.26}$ | $\mathbf{-0.01_{\pm0.02}}$ | $\mathbf{0.63_{\pm0.3}}$ | $0.13_{\pm0.01}$ | $1.58_{\pm0.02}$ | $0.06_{\pm0.05}$ | $2.13_{\pm1.19}$ | $-0.01_{\pm0.09}$ | $2.29_{\pm2.07}$ | $0.37_{\pm0.11}$ | $12.51_{\pm8.54}$ | $-0.07_{\pm0.06}$ | $1.59_{\pm1.1}$ |
| CarCircle1 | $0.72_{\pm0.01}$ | $4.39_{\pm0.1}$ | $0.37_{\pm0.1}$ | $1.38_{\pm0.44}$ | $0.6_{\pm0.01}$ | $1.73_{\pm0.04}$ | $0.73_{\pm0.02}$ | $5.25_{\pm2.76}$ | $0.76_{\pm0.04}$ | $5.46_{\pm2.62}$ | $0.02_{\pm0.15}$ | $2.29_{\pm2.13}$ | $0.7_{\pm0.02}$ | $5.72_{\pm3.04}$ |
| CarCircle2 | $0.76_{\pm0.03}$ | $6.44_{\pm0.19}$ | $0.54_{\pm0.08}$ | $3.38_{\pm1.3}$ | $0.66_{\pm0.0}$ | $2.53_{\pm0.03}$ | $0.72_{\pm0.04}$ | $6.58_{\pm3.02}$ | $0.74_{\pm0.04}$ | $6.82_{\pm2.95}$ | $0.44_{\pm0.1}$ | $2.69_{\pm2.61}$ | $0.77_{\pm0.03}$ | $7.99_{\pm4.23}$ |
| CarGoal1 | $0.39_{\pm0.04}$ | $0.33_{\pm0.12}$ | $0.24_{\pm0.08}$ | $0.28_{\pm0.11}$ | $0.66_{\pm0.01}$ | $1.21_{\pm0.17}$ | $\mathbf{0.47_{\pm0.05}}$ | $\mathbf{0.78_{\pm0.5}}$ | $0.61_{\pm0.04}$ | $1.13_{\pm0.61}$ | $0.79_{\pm0.07}$ | $1.42_{\pm0.81}$ | $\mathbf{0.35_{\pm0.06}}$ | $\mathbf{0.54_{\pm0.33}}$ |
| CarGoal2 | $0.23_{\pm0.02}$ | $1.05_{\pm0.07}$ | $0.14_{\pm0.05}$ | $0.51_{\pm0.26}$ | $0.48_{\pm0.01}$ | $1.25_{\pm0.14}$ | $0.3_{\pm0.05}$ | $1.44_{\pm0.99}$ | $0.28_{\pm0.04}$ | $1.01_{\pm0.62}$ | $0.65_{\pm0.2}$ | $3.75_{\pm2.0}$ | $\mathbf{0.25_{\pm0.04}}$ | $\mathbf{0.91_{\pm0.41}}$ |
| CarPush1 | $0.22_{\pm0.04}$ | $0.36_{\pm0.12}$ | $0.14_{\pm0.03}$ | $0.33_{\pm0.23}$ | $\mathbf{0.31_{\pm0.01}}$ | $\mathbf{0.4_{\pm0.1}}$ | $0.23_{\pm0.03}$ | $0.43_{\pm0.19}$ | $\mathbf{0.21_{\pm0.02}}$ | $\mathbf{0.54_{\pm0.28}}$ | $-0.03_{\pm0.24}$ | $0.95_{\pm0.53}$ | $\mathbf{0.23_{\pm0.04}}$ | $\mathbf{0.5_{\pm0.4}}$ |
| CarPush2 | $0.14_{\pm0.03}$ | $0.9_{\pm0.08}$ | $0.05_{\pm0.02}$ | $0.45_{\pm0.19}$ | $0.19_{\pm0.01}$ | $1.3_{\pm0.16}$ | $0.15_{\pm0.02}$ | $1.38_{\pm0.68}$ | $0.1_{\pm0.02}$ | $1.2_{\pm0.98}$ | $0.24_{\pm0.06}$ | $4.25_{\pm2.44}$ | $0.09_{\pm0.02}$ | $1.07_{\pm0.69}$ |
| SwimmerVelocity | $0.49_{\pm0.27}$ | $4.72_{\pm4.01}$ | $0.51_{\pm0.2}$ | $1.07_{\pm0.07}$ | $\mathbf{0.66_{\pm0.01}}$ | $\mathbf{0.96_{\pm0.08}}$ | $0.48_{\pm0.33}$ | $6.58_{\pm3.95}$ | $0.3_{\pm0.01}$ | $2.33_{\pm0.04}$ | $0.13_{\pm0.06}$ | $2.66_{\pm0.96}$ | $0.63_{\pm0.06}$ | $7.58_{\pm1.77}$ |
| HopperVelocity | $0.65_{\pm0.01}$ | $6.39_{\pm0.88}$ | $0.36_{\pm0.13}$ | $0.67_{\pm0.27}$ | $\mathbf{0.63_{\pm0.06}}$ | $\mathbf{0.61_{\pm0.08}}$ | $0.78_{\pm0.09}$ | $5.02_{\pm3.43}$ | $0.34_{\pm0.06}$ | $5.86_{\pm4.05}$ | $0.14_{\pm0.09}$ | $2.11_{\pm2.29}$ | $0.13_{\pm0.06}$ | $1.51_{\pm1.54}$ |
| HalfCheetahVelocity | $0.97_{\pm0.02}$ | $13.1_{\pm8.3}$ | $\mathbf{0.88_{\pm0.03}}$ | $\mathbf{0.54_{\pm0.63}}$ | $\mathbf{1.0_{\pm0.01}}$ | $\mathbf{0.01_{\pm0.01}}$ | $1.05_{\pm0.07}$ | $18.21_{\pm8.29}$ | $0.98_{\pm0.03}$ | $6.58_{\pm1.03}$ | $\mathbf{0.29_{\pm0.14}}$ | $\mathbf{0.74_{\pm0.19}}$ | $\mathbf{0.65_{\pm0.01}}$ | $\mathbf{0.0_{\pm0.0}}$ |
| Walker2dVelocity | $0.79_{\pm0.28}$ | $3.88_{\pm3.38}$ | $\mathbf{0.79_{\pm0.05}}$ | $\mathbf{0.04_{\pm0.32}}$ | $0.78_{\pm0.09}$ | $0.06_{\pm0.34}$ | $0.79_{\pm0.01}$ | $0.17_{\pm0.06}$ | $0.86_{\pm0.01}$ | $3.1_{\pm0.65}$ | $0.04_{\pm0.05}$ | $0.21_{\pm0.09}$ | $0.12_{\pm0.01}$ | $0.74_{\pm0.07}$ |
| AntVelocity | $0.98_{\pm0.01}$ | $3.72_{\pm1.46}$ | $\mathbf{0.98_{\pm0.01}}$ | $\mathbf{0.29_{\pm0.1}}$ | $0.98_{\pm0.0}$ | $0.39_{\pm0.12}$ | $1.02_{\pm0.01}$ | $4.15_{\pm1.63}$ | $-1.01_{\pm0.0}$ | $0.0_{\pm0.0}$ | $-1.01_{\pm0.0}$ | $0.0_{\pm0.0}$ | $1.0_{\pm0.0}$ | $3.28_{\pm2.01}$ |
| **SafetyGym Average** | $0.46_{\pm0.35}$ | $3.03_{\pm5.32}$ | $\mathbf{0.34_{\pm0.31}}$ | $\mathbf{0.75_{\pm0.8}}$ | $0.54_{\pm0.21}$ | $1.06_{\pm0.59}$ | $0.5_{\pm0.32}$ | $3.29_{\pm4.97}$ | $0.39_{\pm0.43}$ | $2.7_{\pm3.39}$ | $0.27_{\pm0.37}$ | $2.79_{\pm3.86}$ | $0.37_{\pm0.32}$ | $2.65_{\pm3.24}$ |
| BallRun | $0.6_{\pm0.1}$ | $5.08_{\pm0.74}$ | $0.27_{\pm0.14}$ | $1.46_{\pm0.39}$ | $0.39_{\pm0.09}$ | $1.16_{\pm0.19}$ | $0.76_{\pm0.01}$ | $3.91_{\pm0.35}$ | $-0.47_{\pm0.0}$ | $5.03_{\pm0.0}$ | $0.22_{\pm0.0}$ | $1.27_{\pm0.12}$ | $0.59_{\pm0.0}$ | $3.52_{\pm0.0}$ |
| CarRun | $0.97_{\pm0.02}$ | $0.33_{\pm0.05}$ | $\mathbf{0.94_{\pm0.0}}$ | $\mathbf{0.22_{\pm0.02}}$ | $\mathbf{0.99_{\pm0.01}}$ | $\mathbf{0.65_{\pm0.31}}$ | $\mathbf{0.94_{\pm0.01}}$ | $\mathbf{0.15_{\pm0.91}}$ | $0.68_{\pm0.01}$ | $7.78_{\pm0.09}$ | $0.95_{\pm0.01}$ | $1.79_{\pm0.18}$ | $\mathbf{0.87_{\pm0.0}}$ | $\mathbf{0.0_{\pm0.0}}$ |
| DroneRun | $0.24_{\pm0.02}$ | $2.13_{\pm0.62}$ | $0.28_{\pm0.25}$ | $0.74_{\pm0.97}$ | $\mathbf{0.63_{\pm0.04}}$ | $\mathbf{0.79_{\pm0.68}}$ | $0.72_{\pm0.12}$ | $5.54_{\pm0.81}$ | $0.42_{\pm0.1}$ | $2.47_{\pm0.34}$ | $0.33_{\pm0.1}$ | $3.52_{\pm0.58}$ | $0.67_{\pm0.02}$ | $4.15_{\pm0.1}$ |
| AntRun | $0.72_{\pm0.06}$ | $2.93_{\pm2.4}$ | $0.65_{\pm0.15}$ | $1.09_{\pm0.84}$ | $\mathbf{0.72_{\pm0.04}}$ | $\mathbf{0.91_{\pm0.42}}$ | $0.76_{\pm0.07}$ | $5.11_{\pm2.39}$ | $\mathbf{0.15_{\pm0.02}}$ | $\mathbf{0.73_{\pm0.07}}$ | $\mathbf{0.03_{\pm0.02}}$ | $\mathbf{0.02_{\pm0.09}}$ | $\mathbf{0.61_{\pm0.01}}$ | $\mathbf{0.94_{\pm0.69}}$ |
| BallCircle | $0.74_{\pm0.15}$ | $4.71_{\pm1.79}$ | $0.52_{\pm0.08}$ | $0.65_{\pm0.17}$ | $0.77_{\pm0.06}$ | $1.07_{\pm0.27}$ | $0.69_{\pm0.11}$ | $2.36_{\pm1.04}$ | $0.86_{\pm0.18}$ | $3.09_{\pm1.53}$ | $\mathbf{0.64_{\pm0.01}}$ | $\mathbf{0.76_{\pm0.0}}$ | $0.7_{\pm0.04}$ | $2.61_{\pm0.79}$ |
| CarCircle | $0.58_{\pm0.25}$ | $3.74_{\pm2.2}$ | $0.5_{\pm0.22}$ | $0.84_{\pm0.67}$ | $0.75_{\pm0.06}$ | $0.95_{\pm0.61}$ | $0.63_{\pm0.19}$ | $1.89_{\pm1.37}$ | $0.74_{\pm0.1}$ | $2.18_{\pm1.33}$ | $\mathbf{0.71_{\pm0.02}}$ | $\mathbf{0.33_{\pm0.0}}$ | $0.49_{\pm0.05}$ | $3.14_{\pm2.98}$ |
| DroneCircle | $0.72_{\pm0.04}$ | $3.03_{\pm0.29}$ | $0.56_{\pm0.18}$ | $0.57_{\pm0.27}$ | $0.63_{\pm0.07}$ | $0.98_{\pm0.27}$ | $0.8_{\pm0.07}$ | $3.07_{\pm0.89}$ | $0.78_{\pm0.04}$ | $3.68_{\pm0.44}$ | $-0.22_{\pm0.05}$ | $1.28_{\pm0.97}$ | $0.26_{\pm0.03}$ | $1.02_{\pm0.46}$ |
| AntCircle | $0.58_{\pm0.19}$ | $4.9_{\pm3.55}$ | $0.4_{\pm0.16}$ | $0.96_{\pm2.67}$ | $0.54_{\pm0.2}$ | $1.78_{\pm4.33}$ | $0.58_{\pm0.25}$ | $2.87_{\pm3.08}$ | $0.65_{\pm0.2}$ | $5.48_{\pm3.33}$ | $0.0_{\pm0.0}$ | $0.0_{\pm0.0}$ | $0.17_{\pm0.1}$ | $5.04_{\pm6.74}$ |
| **BulletGym Average** | $0.64_{\pm0.25}$ | $3.36_{\pm3.31}$ | $\mathbf{0.52_{\pm0.27}}$ | $\mathbf{0.82_{\pm1.27}}$ | $0.68_{\pm0.19}$ | $1.04_{\pm1.65}$ | $0.74_{\pm0.25}$ | $3.11_{\pm3.55}$ | $0.48_{\pm0.27}$ | $3.8_{\pm3.95}$ | $0.33_{\pm0.29}$ | $1.12_{\pm1.85}$ | $0.55_{\pm0.24}$ | $2.55_{\pm3.62}$ |
| easysparse | $0.17_{\pm0.05}$ | $1.54_{\pm1.38}$ | $\mathbf{0.11_{\pm0.08}}$ | $\mathbf{0.21_{\pm0.02}}$ | $\mathbf{0.17_{\pm0.14}}$ | $\mathbf{0.23_{\pm0.32}}$ | $0.78_{\pm0.0}$ | $5.01_{\pm0.06}$ | $\mathbf{0.11_{\pm0.0}}$ | $\mathbf{0.86_{\pm0.01}}$ | $-0.06_{\pm0.0}$ | $0.07_{\pm0.02}$ | $0.96_{\pm0.02}$ | $5.44_{\pm0.27}$ |
| eastmean | $0.43_{\pm0.02}$ | $2.82_{\pm0.0}$ | $\mathbf{0.04_{\pm0.03}}$ | $\mathbf{0.29_{\pm0.02}}$ | $\mathbf{0.45_{\pm0.11}}$ | $\mathbf{0.54_{\pm0.55}}$ | $0.71_{\pm0.06}$ | $3.44_{\pm0.35}$ | $\mathbf{0.08_{\pm0.0}}$ | $\mathbf{0.86_{\pm0.01}}$ | $-0.07_{\pm0.0}$ | $0.07_{\pm0.01}$ | $0.66_{\pm0.16}$ | $3.97_{\pm1.47}$ |
| easydense | $0.27_{\pm0.14}$ | $1.94_{\pm1.18}$ | $\mathbf{0.11_{\pm0.07}}$ | $\mathbf{0.14_{\pm0.01}}$ | $\mathbf{0.32_{\pm0.18}}$ | $\mathbf{0.62_{\pm0.43}}$ | $0.26_{\pm0.0}$ | $0.47_{\pm0.01}$ | $\mathbf{0.02_{\pm0.05}}$ | $\mathbf{0.41_{\pm0.22}}$ | $-0.06_{\pm0.0}$ | $0.03_{\pm0.01}$ | $0.5_{\pm0.1}$ | $2.54_{\pm0.53}$ |
| mediumsparse | $0.83_{\pm0.13}$ | $3.34_{\pm0.58}$ | $\mathbf{0.33_{\pm0.34}}$ | $\mathbf{0.3_{\pm0.32}}$ | $0.87_{\pm0.11}$ | $1.1_{\pm0.26}$ | $0.44_{\pm0.0}$ | $1.16_{\pm0.02}$ | $-0.03_{\pm0.0}$ | $0.17_{\pm0.02}$ | $-0.08_{\pm0.02}$ | $0.07_{\pm0.03}$ | $0.71_{\pm0.37}$ | $2.49_{\pm1.9}$ |
| mediummean | $0.77_{\pm0.21}$ | $2.53_{\pm0.83}$ | $0.31_{\pm0.06}$ | $0.21_{\pm0.0}$ | $\mathbf{0.45_{\pm0.39}}$ | $\mathbf{0.75_{\pm0.83}}$ | $0.78_{\pm0.12}$ | $1.53_{\pm0.21}$ | $-0.0_{\pm0.0}$ | $0.34_{\pm0.03}$ | $-0.08_{\pm0.0}$ | $0.05_{\pm0.04}$ | $0.76_{\pm0.34}$ | $2.05_{\pm0.92}$ |
| mediumdense | $0.45_{\pm0.27}$ | $1.47_{\pm1.65}$ | $\mathbf{0.24_{\pm0.0}}$ | $\mathbf{0.17_{\pm0.0}}$ | $0.88_{\pm0.12}$ | $2.41_{\pm0.71}$ | $0.58_{\pm0.21}$ | $1.89_{\pm1.19}$ | $\mathbf{0.01_{\pm0.02}}$ | $\mathbf{0.28_{\pm0.16}}$ | $-0.07_{\pm0.0}$ | $0.07_{\pm0.01}$ | $0.69_{\pm0.13}$ | $2.24_{\pm0.65}$ |
| hardsparse | $0.42_{\pm0.15}$ | $1.8_{\pm1.69}$ | $0.17_{\pm0.05}$ | $3.25_{\pm0.1}$ | $\mathbf{0.25_{\pm0.08}}$ | $\mathbf{0.41_{\pm0.33}}$ | $0.5_{\pm0.04}$ | $1.02_{\pm0.05}$ | $\mathbf{0.01_{\pm0.0}}$ | $\mathbf{0.16_{\pm0.02}}$ | $-0.05_{\pm0.0}$ | $0.06_{\pm0.01}$ | $0.37_{\pm0.1}$ | $2.05_{\pm0.27}$ |
| hardmean | $0.2_{\pm0.17}$ | $1.77_{\pm1.89}$ | $\mathbf{0.13_{\pm0.0}}$ | $\mathbf{0.4_{\pm0.0}}$ | $\mathbf{0.33_{\pm0.21}}$ | $\mathbf{0.97_{\pm0.31}}$ | $0.47_{\pm0.13}$ | $2.56_{\pm0.72}$ | $-0.0_{\pm0.0}$ | $0.21_{\pm0.02}$ | $-0.05_{\pm0.0}$ | $0.06_{\pm0.02}$ | $0.32_{\pm0.19}$ | $2.47_{\pm2.0}$ |
| harddense | $0.2_{\pm0.08}$ | $1.33_{\pm0.87}$ | $\mathbf{0.15_{\pm0.06}}$ | $\mathbf{0.22_{\pm0.01}}$ | $0.08_{\pm0.15}$ | $0.21_{\pm0.42}$ | $0.35_{\pm0.03}$ | $1.4_{\pm0.14}$ | $\mathbf{0.02_{\pm0.0}}$ | $\mathbf{0.26_{\pm0.03}}$ | $-0.04_{\pm0.01}$ | $0.08_{\pm0.01}$ | $0.24_{\pm0.21}$ | $1.68_{\pm2.15}$ |
| **MetaDrive Average** | $0.42_{\pm0.33}$ | $2.06_{\pm1.63}$ | $\mathbf{0.18_{\pm0.27}}$ | $\mathbf{0.58_{\pm0.35}}$ | $\mathbf{0.42_{\pm0.31}}$ | $\mathbf{0.8_{\pm0.61}}$ | $0.54_{\pm0.35}$ | $2.05_{\pm2.7}$ | $\mathbf{0.02_{\pm0.09}}$ | $\mathbf{0.39_{\pm0.52}}$ | $-0.06_{\pm0.01}$ | $0.06_{\pm0.04}$ | $0.58_{\pm0.32}$ | $2.77_{\pm2.87}$ |

Table 3: Evaluation results in terms of the normalized reward and cost. The cost threshold is 1. The ↑ symbol denotes that the higher reward, the better. The ↓ symbol denotes that the lower cost (up to threshold 1), the better. Each value is reported as: mean ± standard deviation over 180 episodes (3 seeds × 3 cost thresholds × 20 evaluation episodes). **Bold**: Safe agents whose normalized cost is smaller than 1. Gray: Unsafe agents. Blue: Safe agent with the highest reward.

## 5.2 Main Results

We present the full experiment results on the 38 datasets in Table 3. Each value is averaged over 3 distinct cost thresholds, 3 random seeds, and the final checkpoint. See Table 5 in the

Appendix A for details about the hyperparameters for each method. Here, **BC-All** refers to behavior cloning trained with all datasets, while **BC-Safe** refers to behavior cloning trained exclusively with safe trajectories that satisfy the constraints.

The performance of tested algorithms provides valuable insights into the challenges of offline safe learning. **BC-All** and **BC-Safe**, focusing on imitating policies rather than estimating Q values, exhibit stark differences: BC-All achieves higher rewards but fails on safety constraints; BC-Safe, fed with only safe trajectories, satisfies most safety requirements, although with conservative performances and lower rewards. This comparison underlines the essential trade-offs between safety and utility in offline safe RL, largely dictated by the training dataset used.

**CDT**, through its advanced architecture and effective data utilization, offers a more balanced performance. Despite struggling with complex tasks in high-stochasticity environments, such as `SafetyGym` tasks, CDT generally yields higher rewards while maintaining safety, outperforming BC-Safe in most tasks.

Contrarily, all Q-learning-based algorithms, including **BCQ-Lag**, **BEAR-Lag**, and **CPQ**, as well as **COptiDICE**, display performance inconsistencies, vacillating between excessive conservatism and riskiness. CPQ, for example, obtains high rewards at significant safety compromise in `Button` tasks, while achieving almost zero cost with low rewards in `MetaDrive` tasks.

These inconsistencies expose the key challenge for Q-learning-based approaches in offline safe RL: accurately estimating the safety performance of trained policies. In standard offline RL, minor biases in Q estimation rarely impact overall performance. However, the safety thresholds introduce new dynamics. Under-estimating cost Q values could result in negligible safety penalties, causing overly risky policies, while overestimations can lead to overly conservative behaviors. To tackle this challenge, future research could focus on developing techniques for precise safety performance estimation in offline environments. This is particularly crucial for the application and evolution of Q-learning-based approaches in offline safe RL.

The disparity observed in the performance of learning algorithms across different simulation environments underscores the influence of task definition on algorithmic efficacy. For instance, while numerous methods struggle to maintain safety in Safety Gymnasium's Circle tasks, the same algorithms can maintain safety in BulletSafetyGym's Circle tasks. Interestingly, their task definitions are almost identical. This divergence in outcomes can be attributed to the different time horizons and simulation steps, where SafetyGymnasium tasks typically feature longer time horizons with shorter simulation steps per iteration, while BulletSafetyGym tasks have shorter time horizons, which could potentially facilitate training. Hence, task design crucially impacts algorithmic performance, suggesting that future research should focus on the relationship between task definition, including CMDP design, and algorithm efficacy to foster safer decision-making systems.

## 5.3 Post-process Filters Experiments

Our experiments further delve into a variety of evaluation criteria, as outlined in section 4.3, through different data manipulation filters, notably data density, partial data discarding,

and noise-level manipulation filters. The results are averaged among the corresponding Bullet-Gym and Safety-Gymnasium suites of tasks.

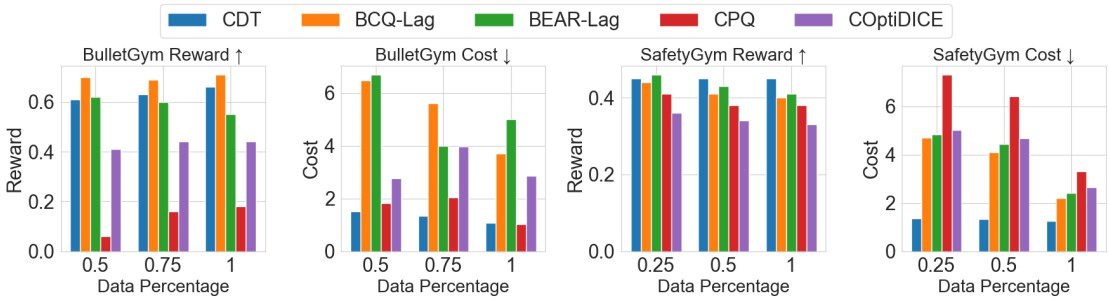

Figure 5: Average performance with different percentage of dataset trajectories.

**Data Density Filter.** As displayed in Figure 5, applying a density filter to vary dataset sizes reveals a trend: most algorithms exhibit a decrease in cost values as more data is employed. This finding underscores the pivotal role of dataset size in influencing algorithmic performance. Interestingly, the safety performance of BCQ-Lag and CPQ are noticeably influenced by data sizes, suggesting that certain algorithms may be more susceptible to data density. In contrast, CDT showcases robustness against sparse data, indicating its potential utility in environments where data collection may be challenging.

**Partial data discarding filter.** The datasets could hardly be perfect and contain all different situations, i.e., with complete coverage of all possible reward and cost returns. The real-world data could be either overly conservative or overly risky. This filter mimics this by selectively discarding trajectories within defined return ranges, which helps us gauge an algorithm's ability to handle unseen safety thresholds and learn from sub-optimal data.

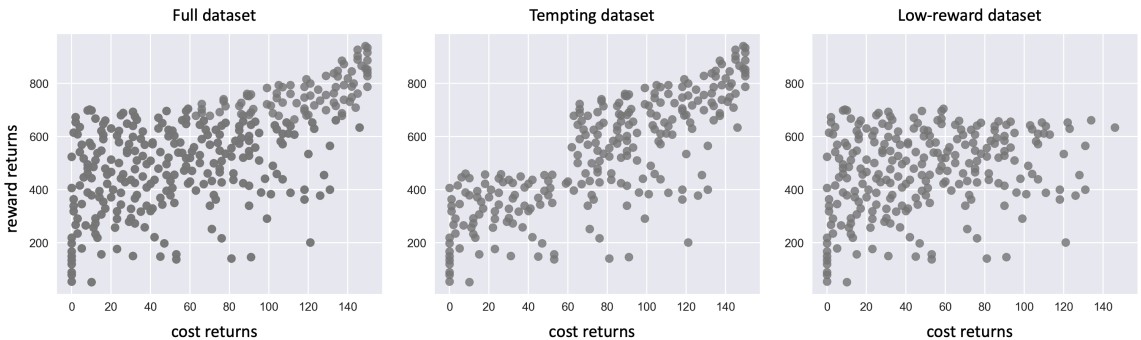

Figure 6: Illustration of the partial data discarding filters.

More specifically, we adopt two data discarding strategies: 1) discarding the top $50\% - 100\%$ reward return trajectories within the $0\% - 50\%$ cost return regions to create a **tempting** dataset (Liu et al., 2022b); 2) discarding the top $50\% - 100\%$ reward return trajectories within the $50\% - 100\%$ cost return regions to create a **low-reward** and sub-optimal dataset. The first dataset is tempting because it contains high-reward and high-cost trajectories that could potentially lead the agent to pursue risky behaviors. The learning algorithm must balance high-reward-high-cost and low-reward-low-cost performance, posing more challenges

than the full datasets. On the contrary, the second **low-reward** dataset could lead to a conservative learned policy. Though the trained agent is safe, the reward could also be low. The tempting dataset and low-reward dataset are shown in Figure 6.

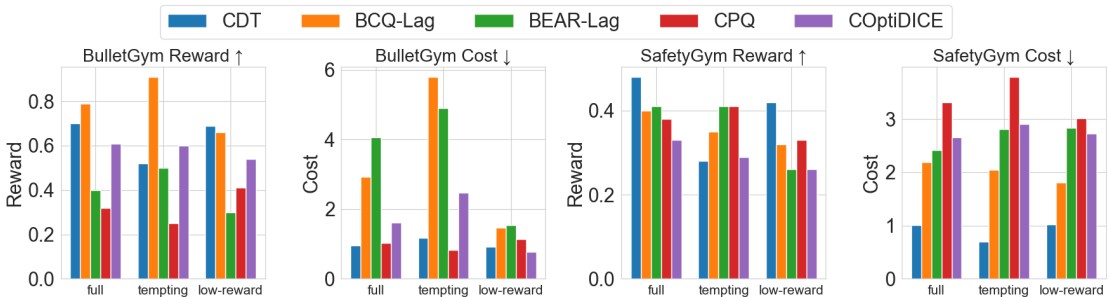

Figure 7: Average performance with different data discarding strategy.

Figure 7 shows the performance under different data-discarding strategies. We can see that The tempting datasets usually lead to high costs and high rewards, i.e., tempting policies, in the BulletSafetyGym tasks. While the low-reward datasets tend to reduce the cost, making the learning algorithm safer. These results consolidate the inherent trade-offs between the reward and cost in the safe learning problem. It also provides an insight that we can adjust the learning difficulty by manipulating the shape of the datasets. Investigating how to selectively use data in the dataset for learning to enhance safety and performance could be an interesting future direction.

**Noise-level manipulation filter.** To evaluate the training robustness of offline learning algorithms, we use this filter to create such datasets that contain different portions ($\alpha\% = 0\%, 1\%, 5\%, 10\%$) of outlier trajectories. Figure. 8 illustrates the performance impact of the noise data filter, which increases the percentage of outlier trajectories. Several algorithms, such as CPQ in SafetyGymnasium tasks, exhibit drastic performance degradation with a substantial reward reduction and cost increase. This phenomenon underscores the critical importance of outlier sensitivity in the evaluation of offline safe RL algorithms. Excessive sensitivity to outliers may introduce instability during the learning process, leading to suboptimal safety and performance outcomes.

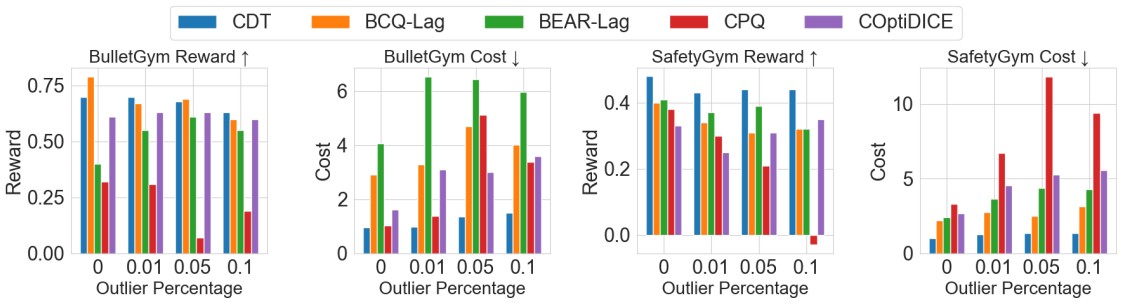

Figure 8: Average performance with different percentages of outlier trajectories.

These experiments offer insights into the robustness and generalization capability of offline safe RL algorithms in the face of dataset variations. From a broader perspective, these

findings illuminate the intricate dynamics and potential difficulties inherent in offline safe RL. The challenge lies in maintaining a delicate balance between reward optimization and safety assurance. Algorithms' performance on this front further underscores the complexity of the datasets used, emphasizing the need for more research into handling these challenges. Through our benchmarks, we hope to foster a deeper understanding of offline safe RL and to accelerate its real-world applications.

## 6 Challenges and Future Directions

While our benchmark study has sought to cover a comprehensive range of factors in offline safe learning, there are also limitations. For example, our study uses only three prevalent safe RL environments. While they provide a broad context for algorithm evaluation, we encourage the community to contribute more datasets to this field. There are also many challenges in RL safety that remain untouched. Below we outline some open problems and potential future directions.

1. **Performance Metrics and Benchmarks**: While we have provided an evaluation framework, there is still room for more sophisticated performance metrics. These could account for various real-world factors, such as environmental changes (Chen et al., 2021a), adversarial attacks (Liu et al., 2022b), and more.

2. **Interpretability and Safety Certification**: As these algorithms become more complex, the need for explainability and theoretical safety guarantees become more pressing (Amodei et al., 2016; Verma et al., 2018; Luo and Ma, 2021). Ensuring that the actions and decisions of these algorithms can be understood by humans and can be certified to be safe will be crucial for their wider acceptance and adoption.

3. **Data Efficiency**: Offline safe RL algorithms are trained from datasets, making data efficiency a critical aspect (Schwarzer et al., 2021). Future research could focus on improving data utilization, possibly through advances in sample-efficient learning techniques or data augmentation strategies (Sinha et al., 2022; As et al., 2022).

4. **Few-shot Online Adaptation**: Offline pretraining plus online adaptation is becoming a popular training paradigm in wide domains (Kumar et al., 2022; Radosavovic et al., 2023). Therefore, the ability of offline safe learning algorithms to adapt to new environments safely with few shot samples is an area ripe for research (Zhu et al., 2020).

5. **Versatility**: The capacity of an algorithm to adapt to varying safety constraints without the need for substantial re-tuning or re-training is pivotal. Currently, only sequential-modeling-based methods (Liu et al., 2023b; Zhang et al., 2023) can effectively achieve this, but there is substantial room for improvement.

6. **Ethics and Fairness**: As offline learning is increasingly deployed in sensitive areas such as healthcare, considerations around ethics and fairness become particularly important (Jabbari et al., 2017; Thomas et al., 2019; Deng et al., 2022). Future work could focus on integrating these factors into the offline safe RL framework.

Addressing these challenges will help drive the field forward, pushing the boundaries of what offline safe RL can achieve. We aspire for the discussions and resources furnished in this work to ignite advancements and foster the evolution of learning-based decision-making systems. Our ultimate goal is to contribute to a future where these systems can be

safely and reliably incorporated into real-world applications, delivering greater efficiency, effectiveness, and most importantly, safety.

## Broader Impact Statement

This paper introduces a benchmarking suite designed for offline safe RL, which is critical in safety-sensitive domains like autonomous driving and robotics. Our benchmark, featuring a data collection pipeline, post-processing filters, and baseline implementations, offers researchers extensive resources to test and refine safe RL algorithms. While we focus on safety, the limitation to three safe RL environments and datasets could inadvertently lead to biases in algorithm development. Researchers must be cautious about overfitting to these environments and remain vigilant about testing their algorithms in diverse, real-world scenarios. Ultimately, We would love to see the offline safe RL applications move from simulated domains to real-word domains, using real-word data. As offline safe RL algorithms become more complex, they might become less interpretable, which could pose challenges in understanding and explaining algorithm decisions. It is important to expand beyond the current scope of environments and datasets to avoid biases and address untouched challenges in RL safety including interpretability, data efficiency, online adaptation, and versatility to facilitate fairness, accountability, and transparency in AI development and deployment.

## Acknowledgments and Disclosure of Funding

The work is partially supported by Google Deepmind with an unrestricted grant. The authors also want to acknowledge the support from the National Science Foundation under grants CNS-2047454.

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

## Appendix A. Benchmark and Dataset Details

### A.1 Hosting, Licensing, and Maintenance Plan

Our dataset and benchmark are accessible through the following URL: `www.offline-saferl.org`. We provide three open-sourced packages, `FSRL` [1] for expert safe RL policies, `DSRL` [2] for managing datasets and environment wrappers, and `OSRL` [3] for offline safe learning algorithms.

The datasets will be hosted on our designated platform accessible via the `DSRL` package. They are also directly downloadable at `http://data.offline-saferl.org/download`. All datasets are licensed under the Creative Commons Attribution 4.0 License (CC BY). As for maintenance, we have established a long-term plan to keep the datasets up-to-date, correct any potential issues, and provide support to users. We also aim to further expand these datasets based on new advances in the field, thus continually promoting progress in offline safe RL research.

The `FSRL` package is under the MIT License, supporting open access and flexibility for modification and reuse. The `DSRL` package and `OSRL` package are licensed under the Apache 2.0 License, following D4RL (Fu et al., 2020) and CORL (Tarasov et al., 2022). This codebase will also be regularly maintained.

### A.2 The FSRL package with safe RL experts for data collection

The **FSRL (Fast Safe Reinforcement Learning)** package provides modularized implementations of safe RL algorithms based on PyTorch (Paszke et al., 2019) and the Tianshou framework (Weng et al., 2021). It offers high-quality implementations of popular safe RL algorithms, serving as an ideal starting point for those looking to explore and experiment in this field. We use this package for dataset collection.

The safe RL algorithms that have been implemented in FSRL are presented in Table 4. They include a first-order method FOCOPS (Zhang et al., 2020), a second-order method CPO (Achiam et al., 2017), Lagrangian-based methods (Stooke et al., 2020), and probabilistic-inference-based method CVPO (Liu et al., 2022a).

| Algorithm | Type | Description |
|---|---|---|
| CPO | On-Policy | Constrained Policy Optimization (Achiam et al., 2017) |
| FOCOPS | On-Policy | First Order Constrained Optimization in Policy Space (Zhang et al., 2020) |
| PPOLagrangian | On-Policy | PPO (Schulman et al., 2017) with PID Lagrangian (Stooke et al., 2020) |
| TRPOLagrangian | On-Policy | TRPO (Schulman et al., 2015) with PID Lagrangian (Stooke et al., 2020) |
| DDPGLagrangian | Off-On-Policy[1] | DDPG (Lillicrap et al., 2015) with PID Lagrangian (Stooke et al., 2020) |
| SACLagrangian | Off-On-Policy[1] | SAC (Haarnoja et al., 2018) with PID Lagrangian (Stooke et al., 2020) |
| CVPO | Off-Policy | Constrained Variational Policy Optimization (Liu et al., 2022a) |

Table 4: Safe RL algorithms implemented in FSRL. [1]Off-On-Policy implies that the base learning algorithm is off-policy, but the Lagrange multiplier is updated in an on-policy fashion.

---

1. `https://github.com/liuzuxin/FSRL`
2. `https://github.com/liuzuxin/DSRL`
3. `https://github.com/liuzuxin/OSRL`

**Dataset collection details.** We collect datasets by training the implemented algorithms with gradually increasing or decreasing cost thresholds for each environment. By varying the algorithm, training hyper-parameters, and threshold, we are able to collect a diverse set of trajectory datasets for each task. These datasets were then merged and applied with a density filter to discard redundant trajectories with high concentrations in the cost-reward return space.

Our dataset collection process was optimized using our carefully tuned hyper-parameters and two strategies: 1) Application of a density filter to the dataset buffer during collection. This step is instrumental in preventing memory overflow. Without the filter, each task will record all collected trajectories, consuming substantial memory and leading to inefficient use of computing resources. 2) Maintenance of a relatively high minimum standard deviation $(e^{-2})$ for stochastic policies to foster exploration. This approach greatly enhances the diversity of the collected datasets. It becomes particularly relevant when cost thresholds are varied during training; an early convergence of the standard deviation may impede exploration and prevent the algorithm from adapting to new thresholds.

Details of the training configurations and hyper-parameters are available in the code.

### A.3 The DSRL package to manage the datasets and filters

The **DSRL (Datasets for Safe Reinforcement Learning)** package serves as an offline safe RL counterpart to the widely-used D4RL. DSRL follows the same usage and API structure as D4RL (Fu et al., 2020), making it easily accessible to researchers already familiar with D4RL. Moreover, it provides clear documentation and examples to guide users. DSRL also ensures scalability, allowing researchers to handle large-scale datasets effectively and customize their own datasets and environments, which is crucial for testing the efficiency and scalability of their algorithms.

We provide an example code to use the dataset:

```python
import gymnasium as gym
import dsrl

env = gym.make('OfflineCarCircle-v0')

# Each task is associated with a dataset with 7 keys:
# [observations, next_observatiosn, actions, rewards,
# costs, terminals, timeouts]
dataset = env.get_dataset()

# An N x obs_dim Numpy array of observations
print(dataset['observations'].shape)

# Apply dataset filters [optional]
# dataset = env.pre_process_data(dataset, filter_cfgs)

# dsrl abides by the OpenAI gym interface
obs, info = env.reset()
obs, reward, terminal, timeout, info = \
    env.step(env.action_space.sample())
cost = info["cost"]
```

Upon running the script, the data will be automatically fetched and stored in the `/home/user/.dsrl/datasets` directory if it does not already exist. This setup follows a similar procedure as seen in D4RL.

### A.4 Hyperparameters for OSRL algorithms

The **OSRL (Offline Safe Reinforcement Learning)** package is a comprehensive library of offline safe RL algorithm implementations. The framework design is inspired by the CORL (Tarasov et al., 2022) and CleanRL (Huang et al., 2022) libraries, which are widely used by offline RL and online RL researchers due to their high-quality and easy-to-follow single-file implementations. For Q-learning-based methods, we use Gaussian policies with mean vectors given as the outputs of neural networks, and with variances that are separate learnable parameters. The policy networks and Q networks for all experiments have two hidden layers with ReLU activation functions. For Lagrangian-based methods, the $K_P, K_I$ and $K_D$ are the PID parameters that control the Lagrangian multiplier. As for CDT, a fixed set of hyperparameters is used across all tasks. The majority of common parameters, including the gradient steps, remain consistent for all the methods employed. Each method is evaluated with three distinct cost thresholds, three random seeds (0, 10, 20), and the final checkpoint in training. The primary hyperparameters employed in the experiments are summarized in Table 5, and more algorithm-specific parameters can be found in the GitHub repository.

| Common Parameters | BulletGym | SafetyGymnasium | MetaDrive | Parameters of CDT | All tasks |
|---|---|---|---|---|---|
| Actor hidden size | [256, 256] for all methods except CDT | | | Number of layers | 3 |
| VAE hidden size | [400, 400] BCQ-Lag, BEAR-Lag, CPQ | | | Number of attention heads | 8 |
| Cost thresholds | [10, 20, 40] | [20, 40, 80] | [10, 20, 40] | Embedding dimension | 128 |
| Gradient steps | 100000 | | 200000 | Batch size | 2048 |
| $[K_P, K_I, K_D]$ | [0.1, 0.003, 0.001] BCQ-Lag, BEAR-Lag | | | Context length K | 300 |
| Batch size | 512 | | | Learning rate | 0.0001 |
| Actor learning rate | 0.0001 | | | Droupout | 0.1 |
| Critic learning rate | 0.001 | | | Adam betas | (0.9, 0.999) |

Table 5: Hyperparameters for `OSRL`

### A.5 Author Responsibility Statement

As the authors, we hereby affirm that we bear full responsibility for the datasets provided in this submission. We confirm that to the best of our knowledge, no rights are violated in the collection, distribution, and use of these datasets. They are provided under the Creative Commons Attribution 4.0 International License, which permits unrestricted use, distribution, and modification, provided appropriate credit is given to the original authors.

## Appendix B. Dataset Documentation and Visualization

### B.1 Dataset Breakdown Details and Intended Uses

The datasets included in this submission are intended for use in the research and development of offline safe learning algorithms. They are diverse, encompassing three different safe RL environments and are designed to test algorithms on a range of safety thresholds. Documentation of the dataset, including a detailed breakdown of environments, tasks, and data sizes, can be found in the following Table 6. The `Max Cost` column denotes the maximum cost return in the dataset trajectories.

| Benchmarks | Task | Max Timestep | Action Space | State Space | Max Cost | Trajectories |
|---|---|---|---|---|---|---|
| SafetyGymnasium | SafetyPointGoal1-v0 | 1000 | 2 | 60 | 100 | 2022 |
| | SafetyPointGoal2-v0 | 1000 | 2 | 60 | 200 | 3442 |
| | SafetyPointButton1-v0 | 1000 | 2 | 76 | 200 | 2268 |
| | SafetyPointButton2-v0 | 1000 | 2 | 76 | 250 | 3288 |
| | SafetyPointPush1-v0 | 1000 | 2 | 76 | 150 | 2379 |
| | SafetyPointPush2-v0 | 1000 | 2 | 76 | 200 | 3242 |
| | SafetyPointCircle1-v0 | 500 | 2 | 28 | 200 | 1098 |
| | SafetyPointCircle2-v0 | 500 | 2 | 28 | 300 | 895 |
| | SafetyCarGoal1-v0 | 1000 | 2 | 72 | 120 | 1671 |
| | SafetyCarGoal2-v0 | 1000 | 2 | 72 | 200 | 4105 |
| | SafetyCarButton1-v0 | 1000 | 2 | 88 | 250 | 2656 |
| | SafetyCarButton2-v0 | 1000 | 2 | 88 | 300 | 3755 |
| | SafetyCarPush1-v0 | 1000 | 2 | 88 | 200 | 2871 |
| | SafetyCarPush2-v0 | 1000 | 2 | 88 | 250 | 4407 |
| | SafetyCarCircle1-v0 | 500 | 2 | 40 | 250 | 1271 |
| | SafetyCarCircle2-v0 | 500 | 2 | 40 | 400 | 940 |
| | SafetySwimmerVelocity-v1 | 1000 | 2 | 8 | 200 | 1686 |
| | SafetyHopperVelocity-v1 | 1000 | 3 | 11 | 250 | 2240 |
| | SafetyHalfCheetahVelocity-v1 | 1000 | 6 | 17 | 250 | 2495 |
| | SafetyWalker2dVelocity-v1 | 1000 | 6 | 17 | 300 | 2729 |
| | SafetyAntVelocity-v1 | 1000 | 8 | 27 | 250 | 2249 |
| BulletSafetyGym | SafetyBallRun-v0 | 100 | 2 | 7 | 80 | 940 |
| | SafetyCarRun-v0 | 200 | 2 | 7 | 40 | 651 |
| | SafetyDroneRun-v0 | 200 | 4 | 17 | 140 | 1990 |
| | SafetyAntRun-v0 | 200 | 8 | 33 | 150 | 1816 |
| | SafetyBallCircle-v0 | 200 | 2 | 8 | 80 | 886 |
| | SafetyCarCircle-v0 | 300 | 2 | 8 | 100 | 1450 |
| | SafetyDroneCircle-v0 | 300 | 4 | 18 | 100 | 1923 |
| | SafetyAntCircle-v0 | 500 | 8 | 34 | 200 | 5728 |
| MetaDrive | SafeMetaDrive-easydense-v0 | 1000 | 2 | 261 | 85 | 1000 |
| | SafeMetaDrive-easysparse-v0 | 1000 | 2 | 261 | 85 | 1000 |
| | SafeMetaDrive-easymean-v0 | 1000 | 2 | 261 | 85 | 1000 |
| | SafeMetaDrive-mediumdense-v0 | 1000 | 2 | 261 | 50 | 1000 |
| | SafeMetaDrive-mediummean-v0 | 1000 | 2 | 261 | 50 | 1000 |
| | SafeMetaDrive-mediumsparse-v0 | 1000 | 2 | 261 | 50 | 1000 |
| | SafeMetaDrive-harddense-v0 | 1000 | 2 | 261 | 85 | 1000 |
| | SafeMetaDrive-hardsparse-v0 | 1000 | 2 | 261 | 85 | 1000 |
| | SafeMetaDrive-hardmean-v0 | 1000 | 2 | 261 | 85 | 1000 |

Table 6: Dataset details. The Max Cost column means the maximum cost return in the dataset trajectories.

## B.2 Dataset cost-reward-return plot visualization.

We visualize the cost-reward-return plot as described in section 3. Each dot is associated with trajectories with corresponding cost and reward returns. Specifically, for each trajectory, we compute its total reward and total cost. Plotting these points on a two-dimensional plane where the x-axis represents the total cost and the y-axis represents the total reward, we obtain a scatter plot that characterizes the trade-offs between reward maximization and constraint satisfaction.

High diversity in a dataset is reflected by a wide spread of points on the cost-reward plot. This implies that the dataset contains trajectories that exhibit various trade-offs between cost and reward, thus providing rich training data for offline safe RL algorithms to learn from. On the contrary, a dataset with low diversity will have points that cluster closely together, indicating limited variety in terms of cost-reward trade-offs. Such a plot serves as an effective and intuitive tool for understanding the properties of offline safe RL datasets. It offers valuable insights into the dataset's composition, revealing the degree of challenge and diversity embedded within. This, in turn, aids in selecting appropriate datasets for benchmarking and comparison of various offline safe RL algorithms.

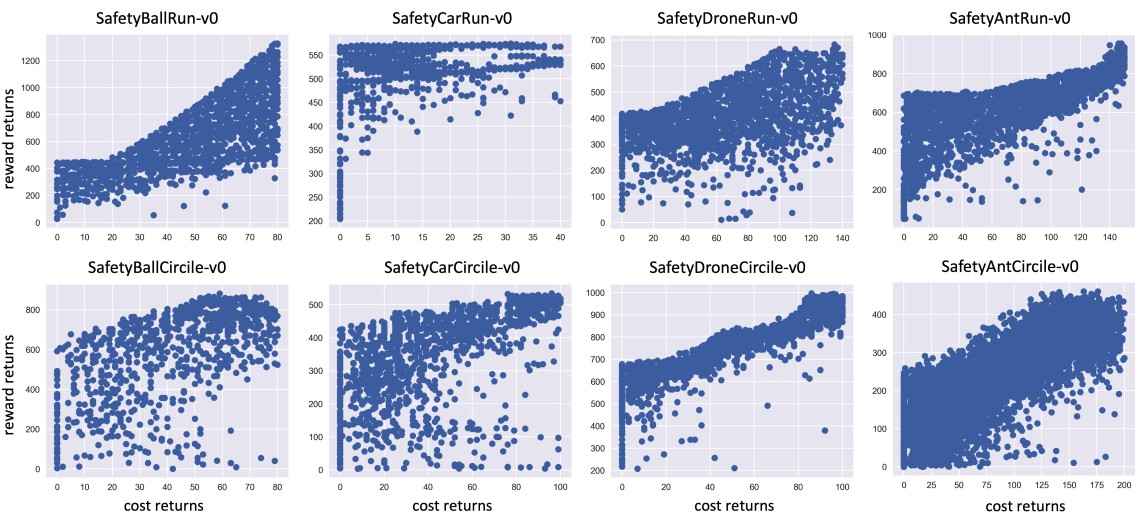

Figure 9: Visualization of BulletSafetyGym dataset trajectories on the cost-reward return space.

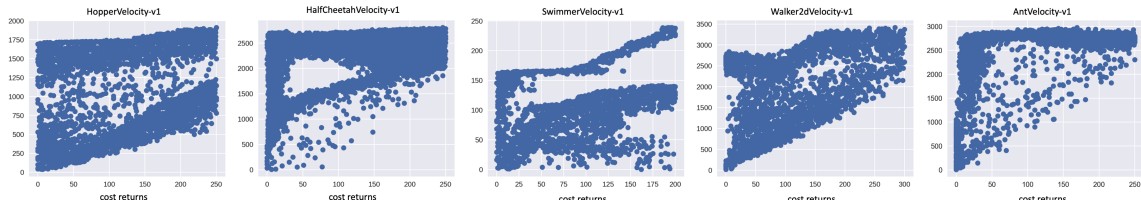

Figure 10: Visualization of Velocity dataset trajectories on the cost-reward return space.

Analyzing the figures provided, we can generally discern an increasing trend for the reward frontiers in relation to the cost returns. In other words, as cost return increases, so too might the reward return, underscoring the inherent trade-off between reward and

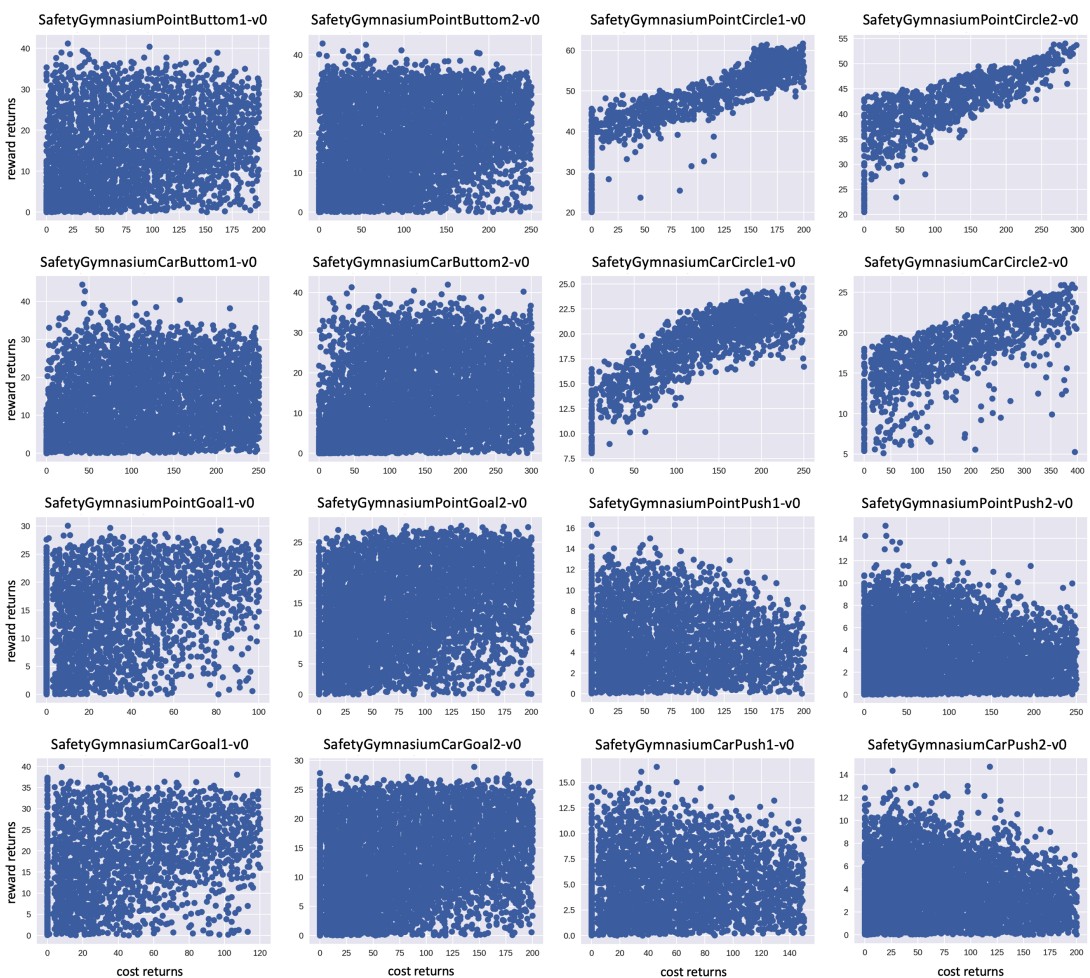

Figure 11: Visualization of SafetyGymnasium dataset trajectories on the cost-reward return space.

cost. This phenomenon aligns with findings discussed in previous works (Liu et al., 2022b, 2023b). It's particularly pronounced in BulletSafetyGym tasks (Figure 9) and the velocity tasks (Figure 10), as these tasks are largely deterministic - their initial states and transition dynamics are not heavily influenced by randomness. We can thus infer that loosening the safety cost threshold may open up opportunities for task utility reward improvement.

In contrast, the same clear increasing trend is not observable in many highly stochastic SafetyGymnasium tasks (Figure 11), such as Goal, Button, and Push. These tasks introduce an element of randomness in the environment, where the initial state is drawn from a random distribution, significantly impacting the final reward and cost. For instance, in the Goal task, random initialization might result in a direct path between the agent's start position and the goal, enabling the completion of the task with zero constraint violations. Consequently, the datasets contain high-reward, low-cost trajectories due to these "lucky" initializations.

For the autonomous driving tasks in MetaDrive (Figure 12), the cost results from three safety-critical scenarios: (i) collision, (ii) out of road, and (iii) over-speed. In this case,

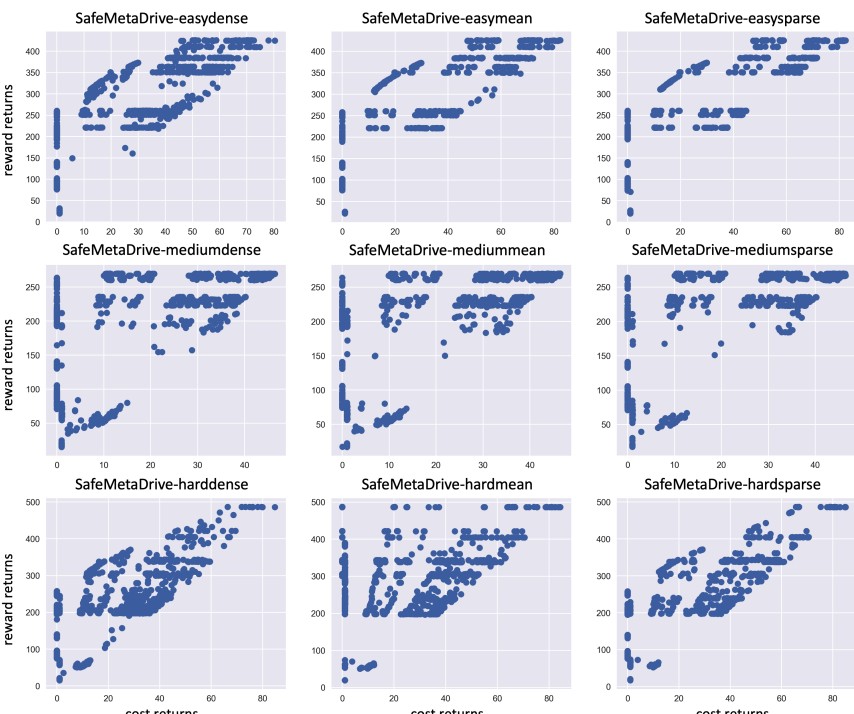

Figure 12: Visualization of MetaDrive dataset trajectories on the cost-reward return space.

the environment's stochasity mainly comes from the random initialization of surrounding traffic flows and the map configuration. To foster the diversity of sampled trajectories within offline datasets, we utilize varying parameters to moderate the aggressiveness of the Intelligent Driver Model (IDM) policies (Kesting et al., 2007) of the ego vehicles. We can also observe an increasing trend for the reward frontiers with respect to the episodic cost returns in most of these environments.

It's worth noting, however, that even though the cost-reward return plot of the dataset might not accurately reflect the reward-cost trade-off, the training curves of the expert policies do display a significant trend. This is because each policy is evaluated on multiple episodes and uses expectations as the evaluation metrics. In other words, under varying cost conditions, the cost value function and the reward value function of the policy can still reflect the trade-offs when considering expectations. This concept is discussed in more detail in (Liu et al., 2022b).

