# OpenReview forum: "Datasets and Benchmarks for Offline Safe Reinforcement Learning"
_DMLR — Accepted by DMLR_

### Review · Reviewer_zsFp · 2023-12-29

**Recommendation:** 4
**Confidence:** 2

**Summary Of Contributions:**

The paper introduces a benchmarking platform for offline safe reinforcement learning (RL), providing a standard testing ground for evaluating and comparing safe RL algorithms.
In offline reinforcement learning, an algorithm is given a dataset that includes multiple trajectories, accompanied by a reward function and cost function, here there are also safe conditions that the algorithms must work by. The paper introduces a collection of post-processing filters to simulate different data collection conditions, resulting in datasets with varying difficulty levels.
The authors also implement a data wrapper and state-of-the-art offline safe learning algorithms, making it a valuable resource for researchers and practitioners in this field.
The paper includes a thorough empirical analysis, utilizing significant computational resources, to gain insights into the strengths and limitations of offline safe RL algorithms and their benchmark. The authors aim to foster collaboration, accelerate innovation, and contribute to the broader adoption of safe RL solutions in safety-critical applications by making their datasets and codebase publicly available.

**Strengths:**

This is a great initiative, contributing a valuable resource for offline RL researchers. The collection of the dataset seems valid, and the analysis of existing algorithms and baselines is also very valuable.

**Audience:**

Yes

**Broader Impact Concerns:**

I have no concerns for the broader impact, and the authors did already add a broader impact statement.

**Claims And Evidence:**

yes

**Datasets And Benchmarks:**

extremely specified and detailed paper about the data collection and benchmark creation, also about ethical considerations etc.

**Extended Submissions:**

not that I know of

**Limitations:**

The authors did refer to a few limitations of this research in their paper, one of them is the small collection of tasks. I agree with the authors that more types of tasks for offline RL could be beneficial. In any case, their methods for creating datasets can also be applied on new tasks.

**Requested Changes:**

I couldn't find anything wrong with the paper.

**Strengths And Weaknesses:**

* The paper is easy to read and follow.
* The empirical analysis is comprehensive and thorough.
* The authors present a valuable resource for offline RL research, incorporating multiple techniques to create datasets and filters.
* The paper presents a new evaluation scheme to evaluate algorithms applied on their benchmark and implements multiple algorithms.
* The analysis of results is clear and presents new findings about algorithm performance in this task, especially the Q-learning based methods.


I could not find any weak points in this research; the limitations section describes excellent points for future work.

---

### Review · Reviewer_Uzp9 · 2024-01-26

**Recommendation:** 4
**Confidence:** 2

**Summary Of Contributions:**

This paper presents an entire suit to benchmark safe offline RL algorithms. The benchmark suit is divided into three parts: Data collection via safe policies in existing safe RL simulation environments (gyms) called FSRL / Dataset unifications and targeted deterministic manipulations in DSRL / evaluations of safe offline RL implementations in OSRL.The authors provide a large rage of implementations for all three packages, as well as some example evaluations of the design choices within all three parts of the pipeline.

**Strengths:**

(see above)

**Audience:**

Yes

**Claims And Evidence:**

Yes. Some improvements needed for automatic evaluation metrics quantifying the impact of control parameters and potentially a change in the normalization process of the cost to ensure unbiased average.

**Datasets And Benchmarks:**

Everything provided in the appendix.

**Extended Submissions:**

N/A

**Limitations:**

(see above)

**Requested Changes:**

The requested changes are grouped by section and pointed to the weaknesses from above.

Section 1:
- A single related work (D4RL) is mentioned without mentioning of others. This could mislead the reader to think there is only 1 related work.
- The mentioned API of D4RL could be included for full self-containment and clarity of reading the paper

Section 3:
- Include distribution statistics (which are anyway used for the evaluation metrics, e.g., min and max along axes) into the process. More complex statistics, such as density, could mentioned.

Section 4:
- The statistics from Section 3 would also help quantify the term 'broad spectrum of datasets' FSRL generated.
- Do not call it 'training tricks', but rather 'settings' or 'best-practices'
- In general, the transformations change the underlying data distribution to provide better generalization or robustness against outliers. Point to related work in the broad field of ML (e.g., early work on augmentation or data programming)
- The legend in Figure 4 is confusing, place it outside of the subfigures
- (4.5) Provide a visualization of the metrics in the cost-reward plots.
- Is the normalized reward for any target threshold?
- The normalized reward will always be bound between 0 and 100. The same does not hold true for the cost. This difference in scale between dataset for evaluation can skew the average cost towards high regret datasets and thus not offer a meaningful number.

Section 5:
- Add difficulty into the table (to help understand the statement 'despite struggling with complex tasks in high-stochasticity environments'
- The average cost in all Figure 5 bars are much higher than Table 3. Is this an artefact of having the cost not being upper bounded?
- (5.3) why is meta-drive not evaluated here?
- The impact of data discarding in Figure 7 (right most) is not visible. The evaluation would benefit from automatic numbers to quantify the impact of control parameters such as discarding data and how much the reward and cost changed, and how often the strategy resulted in safe tragectories.
- The protocol of how to do noise manipulation should be move to section 4. Only the hyper paremeters (what alphas to use) should be in this section.

**Strengths And Weaknesses:**

Strengths:
- S1: Typically, ML algorithms are evaluated for a fixed range of pre-defined dataset. By integrating the data collection and deterministic transformation aspects into the benchmark suit, it enables a wider range of targeted analysis
- S2: The cos-reward plots introduced are intuitive and easy to understand.
- S3: The paper is self-contained and easy to follow

Weaknesses

- W1: The full potential of the benchmark is not clearly highlighted, An overview of the control parameters one can impact within the benchmark suite would be useful. For instance, once data is collected in FSRL, it is considered fixed for the experiments provided in the paper. Once could nonetheless use the modularity of the benchmark to evaluate some hypotheses the author (e.g., impact of longer time horizons) by fixing the transformations and algorithms, and only changing this parameter in the data collection pipeline (if the simulation environments allow for it).
- W2: The evaluation results lack automatic numbers to quantify the impact of control parameters and rely solely on visual analysis.
- W3: There are some concerns about the evaluation metric, specifically averaging the (normalized) cost in the evaluation without having normalized the costs such that they have a unified range across datasets (see requested changes). This can negatively impact the comparison with manipulated datasets if the trajectory results in outliers.
- W4: I wonder whether there would be a more formal way to characterise dataset statistics beyond the visualized cost-reward plots? If some distribution statistics were introduced and captured, this would further strengthen the wide-range applicability of the benchmark suit.

---

### Review · Reviewer_D54L · 2024-01-30

**Recommendation:** 4
**Confidence:** 2

**Summary Of Contributions:**

The paper proposes a novel benchmark for evaluating reinforcement learning (RL) algorithms that are trained in an offline fashion and with a strict focus on safety. Safety is achieved by performing actions that not only aim to maximize rewards (standard RL) but also minimize costs incurred by violations of safety constraints. The authors argue that no benchmark exists for this highly specific yet important scenario and they provide the first benchmark to fill this gap. Their benchmarking framework consists of three packages: (1) FSRL which runs online RL simulations and collects data from the agent; (2) DSRL which takes the collected data and performs transformations and various filters; and (3) OSRL which contains implementations of offline safe learning algorithms along with mechanisms for evaluating them using the collected and processed datasets. Finally, the authors evaluate several offline safe RL algorithms on the datasets that they produced and provide several insights about the current state of the art.

**Strengths:**

**(S1)** The authors have put together a solid benchmark with a decent number of tasks and a comprehensive framework both for evaluating RL models and for generating more benchmark data.

**(S2)** The paper is relevant as it addresses a gap in benchmarks for offline safe RL -- an area with several available methods.

**(S3)** The data processing filters that authors include in the benchmark are an interesting way to draw deeper insights into the performance of RL methods.

**(S4)** The authors spend a full page describing the limitations of their work and discussing opportunities for future work.

**Audience:**

Yes

**Broader Impact Concerns:**

The authors provide a broader impact statement and I see no concerns with it.

**Claims And Evidence:**

N/A

**Datasets And Benchmarks:**

Some details about the data organization are presented in the appendix with additional links to a website and a code repository.

**Extended Submissions:**

The submission does not appear to be extended.

**Limitations:**

**(L1)** The authors could have done a better job introducing and motivating the specific problem they target. I feel that its significance is not self-evident, especially to the broader DMLR audience. The authors argue that offline RL and safe learning are important due to a lot of existing work on the topic (they cite notable examples). However, this argument can be made much stronger by giving us concrete scenarios and examples where offline safe RL is absolutely necessary and where not having it results in a significant burden.

**(L2)** The authors characterize trajectories according to their total cost and total reward, and they claim that it is beneficial to have datasets that exhibit diversity in these two dimensions. They only briefly explain this in section 3.2, but to me, this feels insufficient. Namely, if two trajectories with different total costs and total rewards are considered diverse, then I presume that two trajectories with the same total cost and total reward would be considered "similar". However, it is not self-evident to me why this is the case. I could imagine vastly different trajectories with the same reward and cost, and both being important to expose our RL agent to. Hence, a discussion on this topic would be beneficial, preferably with concrete examples that would give us some intuition to interpret the plots in Figure 2.

**(L3)** In Section 4.1 the authors give a brief description of the tasks that they focus on. It would be beneficial if they could briefly provide some concrete examples of tasks in this section, just to get the reader a feeling about the specific types of tasks that are featured in the benchmark.

**(L4)** The authors measure the quality of each algorithm by running it with three distinct target constraint thresholds and then averaging them. It is unclear why this metric is the most relevant one. The authors compare it to ROC AUC which feels like a stretch since ROC has a probabilistic interpretation which makes it a good measure. I think the paper would benefit from a discussion that would provide a clear motivation for the type of measurement that is used. Since this is a benchmark paper, the choice of measurement feels like a fundamental one.

**Requested Changes:**

**(C1)** Section 1 could include more concrete arguments that should motivate the need for safe offline RL with specific examples that describe cases where not having safe RL is problematic. (See L1)

**(C2)** My understanding is that the OSRL involves a framework for offline learning and online evaluation. If true, it would be good to make this explicit.

**(C3)** Section 3.2 could be improved by introducing a discussion about why the two dimensions (total reward and total cost) are fundamental for characterizing trajectories. (See L2)

**(C4)** Section 4.1 states "The full procedure is visualized in Figure 2". This figure contains 5 scatter plots of reward/cost characteristics and doesn't obviously resemble a visualization of a procedure. I recommend either changing the statement or adding a real visualization of the filtering procedure.

**(C5)** Section 4.1 could benefit from adding some examples of specific tasks (see L3)

**(C6)** Section 4.4 mentions the "type row" in Table 2, but I see only the "Algorithm" and "Base Method" rows. Perhaps this is a typo?

**(C7)** Section 5.1 would benefit from explaining why that specific measure of model quality is selected (i.e. measuring with three target thresholds and averaging). After reading this part, many questions pop up -- How are the thresholds selected? Why not more thresholds? Why is this fundamental? (See L4)

**Strengths And Weaknesses:**

N/A